## Mechanistic investigations of the formation of highly oxidized

## 2 products from the multi-generation OH oxidation of styrene

| 3        | Long Chen, <sup>1,2,3</sup> Yu Huang, *, <sup>1,2,3</sup> Yonggang Xue, <sup>1,2,3</sup> Long Cui, <sup>1,2,3</sup> Zhihui Jia <sup>4</sup> |
|----------|---------------------------------------------------------------------------------------------------------------------------------------------|
| 4        | <sup>1</sup> State Key Laboratory of Loess Science, Institute of Earth Environment, Chinese                                                 |
| 5        | Academy of Sciences, Xi'an 710061, China                                                                                                    |
| 6        | <sup>2</sup> National Observation and Research Station of Regional Ecological Environment                                                   |
| 7        | Change and Comprehensive Management in the Guanzhong Plain, Xi'an 710061,                                                                   |
| 8        | China                                                                                                                                       |
| 9        | <sup>3</sup> Shaanxi Key Laboratory of Atmospheric and Haze-fog Pollution Prevention, Xi'an                                                 |
| 10       | 710061, China                                                                                                                               |
| 11       | <sup>4</sup> School of Materials Science and Engineering, Shaanxi Normal University, Xi'an,                                                 |
| 12       | Shaanxi, 710119, China                                                                                                                      |
| 13       |                                                                                                                                             |
| 14       |                                                                                                                                             |
| 15       |                                                                                                                                             |
| 16       |                                                                                                                                             |
| 17       |                                                                                                                                             |
| 18       |                                                                                                                                             |
| 19       |                                                                                                                                             |
| 20       |                                                                                                                                             |
| 21       | *Corresponding author:                                                                                                                      |
| 22<br>23 | Prof. Yu Huang, E-mail address: <a href="mailto:huangyu@ieecas.cn">huangyu@ieecas.cn</a>                                                    |

#### Abstract

24

Styrene is the second most efficient aromatic compound in the formation of 25 secondary organic aerosol (SOA) after toluene. Recent experiments have identified C<sub>7</sub> 26 27 and C8 series compounds as the main components of SOA in the photooxidation of styrene. However, their molecular structures and formation pathways remain largely 28 uncharacterized. Herein, the formation mechanisms of highly oxidized products from 29 the multi-generation OH oxidation of styrene are studied using the quantum 30 31 chemistry methods. The calculations show that the multi-generation OH oxidation mechanisms of styrene are modulated by RO2 and RO radicals. For the first 32 generation OH oxidation, the addition of OH radicals to the  $C_{\beta}$ -site of vinyl group is 33 the dominant pathway, and the main C<sub>7</sub>- and C<sub>8</sub>-products are benzaldehyde, 34 1<sup>st</sup>-ROOH (C<sub>8</sub>H<sub>10</sub>O<sub>3</sub>) and 1<sup>st</sup>-RONO<sub>2</sub> (C<sub>8</sub>H<sub>9</sub>NO<sub>3</sub>). For the second generation OH 35 oxidation, OH-addition reaction occurring at the ortho-position of 1st-ROOH and 36 1<sup>st</sup>-RONO<sub>2</sub> has a significant dominance. The peroxide bicyclic peroxy radicals (BPR) 37 can react with  $HO_2$  and NO to form the  $C_8$ -products  $2^{nd}$ -ROOH ( $C_8H_{12}O_8$ ) and 38  $2^{nd}$ -RONO $_2$  ( $C_8H_{10}N_2O_{10}$ ). The formed peroxide bicyclic alkoxy radical (BAR) can 39 proceed either the ring-opening reactions to form the multifunctionalized dicarbonyls 40 and ketene-enols, or the cyclization reactions to generate the highly oxidized 41 C<sub>6</sub>-epoxides with the branching ratios of ~30%. For the third generation OH 42 oxidation, syn-OH-addition occurring at the C=C double bond of 2<sup>nd</sup>-ROOH and 43 2<sup>nd</sup>-RONO<sub>2</sub> has the smallest barrier. The major C<sub>8</sub>-products are the 44 multifunctionalized hydroperoxides and organic nitrates. The volatility of the 45 multi-generation OH oxidation products significantly decreases with increasing the 46 47 number of OH oxidation steps.

## 1. Introduction

Aromatic compounds are recognized as the significant secondary organic aerosol (SOA) precursors, accounting for 20%-30% of the total volatile organic compounds (VOCs) and up to ~60% of the urban atmosphere (Xu et al., 2020; Yan et al., 2019; Yu

al., 2005; Forstner et al., 1997). The primary sources include the incomplete 54 combustion, solvent evaporation, and industrial emission, and the secondary sources 55 56 involve the biofuel and biomass burning (Xu et al., 2020; Cabrera-Perez et al., 2016; Li et al., 2019). The most abundant aromatic compounds, including benzene, toluene, 57 ethylbenzene, xylenes, styrene and trimethylbenzenes, are highly present in urban 58 environments (Cabrera-Perez et al., 2016; Koppmann, 2008). The degradation of 59 aromatic compounds initiated by the atmospheric oxidants (e.g., OH radicals, NO<sub>3</sub> 60 radicals, O<sub>3</sub>, and Cl atom) leads to the formation of the highly oxidized products (e.g., 61 nitroaromatics, dicarbonyls, cresols, epoxides) (Ji et al., 2017; Wu et al., 2014; Fu et 62 al., 2023; Wang and Li, 2021; Wang et al., 2013; Zaytsev et al., 2019; Wang et al., 63 2020), significantly contributing to new particle formation (NPF) and SOA formation 64 (up to 50% in eastern China) in the atmosphere (Wang et al., 2017; Wang et al., 2020; 65 66 Garmash et al., 2020; Molteni et al., 2018; Nie et al., 2022). 67 Styrene has been identified as the second most efficient aromatic compound in the formation of SOA after toluene (Sun et al., 2016; Tajuelo et al., 2019a), which is 68 69 primarily emitted from the anthropogenic activities such as solvent usage and vehicle 70 exhaust (Cho et al., 2014; Wu et al., 2021). Styrene is detected at the ppb levels in 71 urban environments, with the mixing ratios of 0.06-4.50 ppb (Cho et al., 2014; Huang 72 et al., 2019), which has been classified as a hazardous air pollutant in the 1990 Clean Air Act due to the potential mutagen and carcinogen (Environmental Protection 73 Agency (EPA), 1990). Therefore, it is very necessary to investigate the degradation 74 75 mechanisms of styrene under atmospheric conditions. In general, the atmospheric oxidation of styrene initiated by OH radicals is anticipated to be the dominant daytime 76 sink, and the lifetime is estimated to be ~ 8 h under the conditions of typical OH 77 radicals concentrations ([OH] =  $\sim 2 \times 10^6$  molecules cm<sup>-3</sup>) (Wu et al., 2021; Shen et 78 al., 2022). Due to the existence of highly reactive vinyl and aromatic groups, 79 80 OH-initiated oxidation of styrene mainly comprise two kinds of pathways: H-abstraction and OH-addition, in which  $C_{\beta}$ -site OH-addition reaction is expected to 81 be the predominant pathway (Wu et al., 2021; Wang et al., 2015; Zhang et al., 2024). 82

et al., 2022; Cabrera-Perez et al., 2016; Iyer et al., 2023; Wang et al., 2017; Bloss et

The formed products can combine with an O<sub>2</sub> molecule leading to the first generation peroxyl radicals, which can further react with NO resulting in the formation of 84 benzaldehyde and formaldehyde. The barrier of the rate-limiting step is predicted to 85 86 be 28.4 kcal/mol (Wang et al., 2015), implying that benzaldehyde is unlikely to be the sole primary product in the oxidation of styrene due to their higher barriers. 87 Additionally, carbonyl oxides, formed in the ozonolysis of styrene, serve as the chain 88 units participating in the formation of oligomers (Yu et al., 2022). The volatility of 89 oligomers decreases dramatically as the successive addition of carbonyl oxides 90 increases, eventually transforming into extremely low volatility organic compounds 91 (ELVOC) and directly participating in NPF. 92 Experimentally, Cho et al., (2014) investigated the kinetics of the reaction 93 styrene + OH at 240-340 K and 1-3 Torr using the mass spectrometry technique. 94 They found that the addition of OH radicals to the vinyl carbons is dominant, and the 95 determined rate coefficient is  $(5.80 \pm 0.49) \times 10^{-11}$  cm<sup>3</sup> molecule<sup>-1</sup> s<sup>-1</sup> at room 96 temperature. In the smog chamber experiments, Tajuelo et al., (2019a, 2019b and 97 2019c) found that the SOA yields from the photolysis and photooxidation of styrene 98 99 and its homologous species increase with the concentration of initial reactants 100 increasing, and benzaldehyde, benzoyl chloride, acetophenone and formaldehyde are 101 expected to be the primary gas phase products. Yu et al., (2022) investigated the 102 formation of SOA from styrene in an indoor chamber under different NO<sub>x</sub> and RH conditions, and found the SOA yields decrease with increasing RH in both the H<sub>2</sub>O<sub>2</sub> 103 and NO<sub>x</sub> systems. The C<sub>7</sub> and C<sub>8</sub> species are the main products in the H<sub>2</sub>O<sub>2</sub> system, 104 105 while organic nitrates are the major components in the NO<sub>x</sub> system. Although the possible molecular formula and chemical composition of the oxidation products from 106 the reaction styrene + OH are given in the aforementioned studies, the specific 107 molecular structures and formation pathways remain ambiguous. Additionally, to the 108 best of our knowledge, the majority of studies mainly focus on the first generation OH 109 oxidation products to date, while the formation mechanisms of highly oxidized 110 products from the multi-generation OH oxidation of styrene are still limited. 111

113 in the presence of HO<sub>2</sub>/NO are investigated using the quantum chemistry methods. The calculated results arising from the first generation OH oxidation reactions are 114 presented herein for comparison with the available literatures to ascertain the 115 116 reliability of the employed theoretical method. For the multi-generation OH oxidation reactions of styrene, all the possible pathways, including OH-addition, 117 O<sub>2</sub>-addition, cyclization, ring-opening, intramolecular H-shifts, C-C bond scission, 118 and HO<sub>2</sub>-elimination reactions, are taken into account. Additionally, the saturated 119 concentrations of the formed highly oxidized products are estimated to identify their 120 volatility classes. 121

## 2. Computational methods

122

123

126127

139140

## 2.1 Electronic structures and energy calculations

The electronic structures and energy calculations of all stationary points, including reactants (R), intermediates (IM), transition states (TS) and products (P), are performed using the Gaussian 16 program (Frisch et al., 2016). Geometric optimizations of all stationary points on the potential energy surfaces (PESs) are carried out at the M06-2X/6-31+g(d,p) level of theory, since it has reliable performance for describing the noncovalent interactions, thermochemical, and kinetics (Zhao and Truhlar, 2008). Harmonic vibrational frequencies are determined at the M06-2X/6-31+g(d,p) theoretical level to confirm the characteristics of all stationary points (a local minimum or a saddle point). The zero-point vibrational energy (ZPVE) is scaled by a factor of 0.967 (Alecu et al., 2010). Intrinsic reaction coordinate (IRC) calculations are carried out to ascertain the connection of the given TS between the designated local minima R and P (Fukui, 1981). Single point energy calculations are performed at the M06-2X/6-311++G(3df,3pd) level of theory, which has been demonstrated to yield accurate energetics compared to that of the gold standard methods (e.g., QCISD(T), CCSD(T) and ROHF-ROCCSD(T)-F12a) (Ji et al., 2018; Fu et al., 2024 ). The activation energy ( $\Delta E_a$ ) and reaction energy ( $\Delta E_r$ ) are defined as the difference in energy between TS and R, as well as between P and R.

## 2.2 Conformer search

144145

152153

154155

161162

164165

169170

RO<sub>2</sub> radicals, formed in the addition of O<sub>2</sub> to the carbon-centered site of alkyl radicals, have multiple conformers due to the varying attack directions of O<sub>2</sub> (Chen et al., 2021; Fu et al., 2020; Møler et al., 2020). In order to obtain the stable conformer of RO<sub>2</sub> radicals, the conformer search is conducted using the Molclus program (Lu, 2024). For the intramolecular H-shift reactions of RO<sub>2</sub> radicals, the B3LYP/6-31+g(d) optimized structures of R, TS and P are selected as the initial geometries to perform the conformational sampling. Notably, the cyclic TS structure in the conformational sampling of TSs is constrained by adjusting the lengths of the O-O, C-H and O-H bonds in RO<sub>2</sub> radicals. Next, the resulting structures are initially optimized at the same level, since it has been demonstrated to be effective in determining the relative energies of various conformers (Rissanen et al., 2014; Hyttinen et al., 2015). The conformers within 2.0 kcal/mol with respect to the lowest energy conformer are further optimized at the M06-2X/6-31+g(d,p) level of theory. Then, the single point energy calculations are performed at the M06-2X/6-311++G(3df,3pd) level of theory. Similar methodology is adopted to investigate the isomerization reactions of RO radicals.

#### 2.3 Kinetics calculations

The rate coefficients of unimolecular reactions, such as intramolecular H-shift and the cyclization of  $RO_2$  and RO radicals, are calculated using the RRKM theory along with energy-grained master equation (RRKM-ME) (Holbrook et al., 1996), while the rate coefficients of bimolecular reactions, such as OH-addition and  $HO_2$ -elimination, are determined using the traditional transition state theory (TST) (Fernández-Ramos et al., 2007). An asymmetric one-dimensional Eckart model (Eckart, 1930) is taken into consideration for the tunneling correction factors in the rate coefficient calculations of RRKM-ME and TST. A single exponential down model in the RRKM-ME calculations is utilized to approximate the collision transfer ( $<\Delta E>_{\rm down}=200~{\rm cm}^{-1}$ ). The Lennard-Jones parameters of all intermediate species are estimated using the empirical formula as proposed by Gilbert and Smith (1990).

For the intramolecular H-shifts of RO<sub>2</sub> and RO radicals, the rate coefficients are

- computed using the multiconformer transition state theory (MC-TST) (Møler et al.,
- 2016), which is expressed as Eq. (1): (Møller et al., 2016 and 2020; Pasik et al., 2024)

$$k_{\text{MC-TST}} = \kappa \frac{k_{\text{B}}T}{h} \frac{\sum_{i}^{\text{TS conf.}} \exp(\frac{-\Delta E_{i}}{k_{\text{B}}T}) Q_{\text{TS},i}}{\sum_{j}^{\text{R conf.}} \exp(\frac{-\Delta E_{j}}{k_{\text{D}}T}) Q_{\text{R},j}} \exp\left(-\frac{E_{\text{TS}} - E_{\text{R}}}{k_{\text{B}}T}\right)$$
(1)

where  $\kappa$  is the Eckart tunneling coefficient, h is Planck's constant,  $k_B$  is 174 175 Boltzmann's constant, and T is the absolute temperature (298.15 K).  $Q_{TS,i}$  and  $Q_{R,j}$ 176 refer to the partition functions of the corresponding transition state i and reactant j conformers, respectively.  $\Delta E_i$  and  $\Delta E_j$  represent the relative electronic energies 177 178 between the corresponding transition state i and reactant j conformers and the lowest energy conformers, respectively.  $E_{TS}$  and  $E_{R}$  stand for the electronic energies of the 179 180 lowest energy transition state and reactant conformers, respectively. All kinetics calculations are carried out using the KiSThelP 2021 and MESMER 6.0 programs 181 (Glowacki et al., 2012; Canneaux et al., 2013). 182

## 3. Results and discussion

189

192

194

#### 3.1 First generation OH oxidation mechanisms of styrene

Styrene is composed of a benzene ring and a vinyl group, and its oxidation initiated by OH radicals may proceed either on the vinyl group or on the benzene ring. Previous literature has demonstrated that the addition of OH radicals to terminal carbon ( $C_{\beta}$ -site) of a vinyl group in styrene is the dominant pathway, with the branching ratio of 88.2% (Wu et al., 2021). Therefore, the  $C_{\beta}$ -site OH-addition reaction is mainly considered in the present study. Figure 1 depicts that this reaction starts with the formation of a pre-reactive complex IM1, and then transforms into an alkyl radical S1-1 via transition state TS1 with the apparent barrier and activation barrier of -3.5 and 0.8 kcal/mol. The calculated results are in good agreement with the values of -2.6 and 0.6 kcal/mol predicted at the high theoretical level DLPNO-CCSD(T)//M06-2X (Wu et al., 2021; Zhang et al., 2024). The rate coefficient of  $C_{\beta}$ -site OH-addition reaction is estimated to be 1.5 × 10<sup>-11</sup> cm<sup>3</sup> molecule<sup>-1</sup> s<sup>-1</sup> at ambient temperature, which is approximately consistent with the

10<sup>-11</sup> cm<sup>3</sup> molecule<sup>-1</sup> s<sup>-1</sup>) for the total rate coefficient of the reaction styrene + OH (Wu 199 et al., 2021; Zhang et al., 2024)]. The aforementioned results reveal that the 200 201 computational method employed herein is reasonable to describe the atmospheric oxidation mechanisms of styrene. 202 203 Due to the existence of resonance structures with radical character on the aromatic ring, the resulting S1-1 can readily isomerize into three other species, 204 namely, S1-2, S1-3 and S1-4. Next, S1-1 can combine with an O<sub>2</sub> molecule leading to 205 the formation of first generation peroxy radicals S2-1-x ( $\Delta E_r > 59.6$  kcal/mol), which 206 includes eight energetically similar conformers, and the corresponding Boltzmann 207 population is listed in Table S1. Notably, the formations of peroxy radicals S2-2-x 208 from the association reactions S1-2 + O<sub>2</sub> are strongly endothermic ( $\Delta E_r = 8.1-10.4$ 209 kcal/mol), as depicted in Figure S1, suggesting that they have a significant potential to 210 211 redissociate back to S1-2 and O2. The resulting S2-2-x can undergo through various 212 intramolecular H-shifts to yield distinct C-centered and O-centered radicals. Among these competing H-shift pathways, hydrogen transfer from the -OH group to the 213 214 terminal oxygen of -OO group has the lowest barrier ( $\Delta E_a = 17.4 \text{ kcal/mol}$ ). A similar conclusion is also obtained from the association reactions S1-3 +  $O_2$  ( $\Delta E_r = 6.6$ -7.1 215 kcal/mol) and S1-4 + O<sub>2</sub> ( $\Delta E_r = 8.1$ -11.1 kcal/mol) that the formations of RO<sub>2</sub> radicals 216 217 are unfavorable thermochemically (Figures S2 and S3). Therefore, in the present study, we mainly focus on the subsequent reaction mechanisms of peroxy radicals S2-1-x 218 under both low and high NO<sub>x</sub> conditions. 219 220 For the unimolecular reactions of S2-1-x, there are three kinds of pathways. One is the H-shift reactions, where the hydrogen atom migrates from the -CH<sub>2</sub>, -CH and -221 OH groups to the terminal oxygen atom of the -OO group leading to QOOH radicals. 222 Among these competing H-shift reactions, the hydrogen atom transfer from the -OH 223 group to the -OO group proceeds via a five-membered ring transition state to yield a 224 hydroperoxyalkoxy radical S3-1-a, which exhibits the smallest barrier ( $\Delta E_a = 21.0$ 225 kcal/mol). Then, the decomposition of S3-1-a undergoes through the successive 226 elimination of HCHO and an OH radical to generate benzaldehyde. Based on the 227

experimental  $(1.2-6.2 \times 10^{-11} \text{ cm}^3 \text{ molecule}^{-1} \text{ s}^{-1})$  and theoretical values  $(1.7-2.0 \times 10^{-10} \text{ cm}^3 \text{ molecule}^{-1} \text{ s}^{-1})$ 

values of  $\Delta E_a$ , it can be concluded that H-shift reaction is the rate-determining step in the whole process. The other is the intramolecular cyclization, where the -OO group 229 attacks the C=C double bond in the benzene ring forming a cyclic peroxide alkyl 230 231 radical S3-1-c ( $\Delta E_a = 33.6 \text{ kcal/mol}$ ). The last is the HO<sub>2</sub>-elimination reaction, where a concerted process of C<sub>a</sub>-O and C<sub>b</sub>-H bonds scission forms a closed-shell species 232 S3-1-b and a HO<sub>2</sub> radical byproduct ( $\Delta E_a = 33.3$  kcal/mol). The aforementioned 233 results show that the cyclization and HO<sub>2</sub>-eliminaiton reactions are less importance 234 due to their higher barriers. The rate coefficient of the unimolecular reactions of 235 S2-1-x is calculated to be  $1.6 \times 10^{-4}$  s<sup>-1</sup> at room temperature, which is about two 236 orders of magnitude lower than the typical pseudo-first-order rate constants  $k'_{RO2+HO2}$ 237 ( $\sim 0.01 \text{ s}^{-1}$ ) and  $k'_{\text{RO2+NO}}$  ( $\sim 0.01 \text{ s}^{-1}$ ) of the bimolecular reactions of RO<sub>2</sub> radicals with 238 HO<sub>2</sub> radicals and NO (Fu et al., 2024; Pasik et al., 2024). 239 In the presence of NO, the bimolecular reactions of S2-1-x with NO proceed via 240 241 oxygen-to-oxygen coupling to generate organic nitrites ROONO, followed by decomposition to benzaldehyde and CH2OH radical or isomerization to organic 242 nitrates RONO<sub>2</sub>. Wang et al. predicted that the energy barrier of the rate-limiting step 243 244 for the formation of benzaldehyde is 28.4 kcal/mol, which is approximately 4.0 kcal/mol greater than that of the formation of RONO<sub>2</sub> (Wang et al., 2015). Moreover, 245 reactions with HO<sub>2</sub> radicals leading to hydroperoxides ROOH are anticipated to be 246 247 one of the dominant sinks for S2-1-x under low NO<sub>x</sub> conditions. Specifically, SOA yields obtained from the reaction  $RO_2 \cdot + HO_2 \cdot \rightarrow ROOH$  under low  $NO_x$  conditions 248 are greater than those derived from the reaction  $RO_2 \cdot + NO \rightarrow RONO_2$  under high 249 250 NO<sub>x</sub> conditions for some aromatic compounds (e.g, m-xylene, toluene) (Ng et al., 2007). The aforementioned results imply that ROOH and RONO<sub>2</sub> may be significant 251 oxidation products expect benzaldehyde in the photooxidation of styrene. The 252 hypothesis is further confirmed by the recent smog chamber experiment study that C<sub>7</sub> 253 and  $C_8$  series products, as well as organic nitrates are the main components of SOA in 254 the OH-initiated oxidation of styrene under different NO<sub>x</sub> conditions (Yu et al., 2022). 255 Considering that the extensive studies on the oxidation of benzaldehyde have done 256 (Sebbar et al., 2011; Zhao et al., 2022; Iuga et al., 2008), this study primarily focus on 257

the multi-generation OH oxidation mechanisms of ROOH and RONO<sub>2</sub> under different  $NO_x$  conditions.

**Figure 1.** PES for the first-stage oxidation of styrene initiated by OH radicals and the unimolecular reactions of S2-1-x at the M06-2X/6-311++G(3df,3pd)//M06-2X/6-31+g(d,p) level

# 3.2 Second generation OH oxidation mechanisms of 1<sup>st</sup>-ROOH

## and 1<sup>st</sup>-RONO<sub>2</sub>

The first generation products, including hydroperoxides 1<sup>st</sup>-ROOH and organic nitrates 1<sup>st</sup>-RONO<sub>2</sub>, include multiple conformers. To obtain the global minimum of 1<sup>st</sup>-ROOH and 1<sup>st</sup>-RONO<sub>2</sub>, the conformer search is performed by using the Molclus program. The resulting structures are initially optimized at the M06-2X/6-31+g(d,p) level, then the single point energies are calculated at the M06-2X/6-311++G(3df,3pd) level. The global minimum structures of 1<sup>st</sup>-ROOH (S4) and 1<sup>st</sup>-RONO<sub>2</sub> (S5) are presented in Figure S4.

## 3.2.1 The oxidation mechanism of 1st-ROOH initiated by OH

#### radicals

The reaction 1<sup>st</sup>-ROOH (S4) + OH proceeds through the addition of OH radicals to either side of the benzene ring to yield alkyl radicals, as depicted in Figure 2. Based on the values of  $\Delta E_a$ , OH-addition occurring at the opposite direction of –OOH group exhibit significantly lower barriers compared to that at the same direction of –OOH group. This discrepancy in barrier heights could be attributed to the existence of large

the opposite direction of -OOH group, the addition of OH radicals to C<sub>1</sub>-site has the 280 smallest barrier ( $R_{S4-add1}$ ,  $\Delta E_a = 0.8$  kcal/mol) due to the stability of  $P_{S4-add1}$ . The result 281 282 is opposite to Zhang's finding that the addition of OH radicals to C6-site would be the most favorable pathway (Zhang et al., 2024). The difference in the preferred pathways 283 284 can be attributed to the absence of the pre-reactive complexes in Zhang's study, leading to potentially incorrect conclusions. Our result is further supported by the 285 analogous reaction systems, such as the reaction toluene + OH, that the 286 ortho-addition reaction has a significant dominance (Ji et al., 2017; Wu et al., 2014; 287 Zhang et al., 209; Wu et al., 2020). 288  $P_{S4-add1}$  includes two conjugate double bonds ( $C_2=C_3$  and  $C_4=C_5$ ), which can 289 290 readily isomerize to P<sub>S4-add1'</sub> and P<sub>S4-add1'</sub>, as evident from Figure S5. In the present of O2, the attack of an O2 molecule on the C-centered site of PS4-add1, PS4-add1, and 291 292 P<sub>S4-add1</sub>" proceed via the barrierless processes to produce the second generation peroxy 293 radicals P<sub>S4-add1</sub>-OO, P<sub>S4-add1</sub>-OO and P<sub>S4-add1</sub>-OO. The O<sub>2</sub>-addition reaction occurring at the same direction as the -OOH group is defined as syn-O2-addition, while the 294 295 O<sub>2</sub>-addition reaction occurring at the opposite direction as the –OOH group is defined as anti-O<sub>2</sub>-addition. For the reaction  $P_{S4-add1} + O_2 \rightarrow P_{S4-add1}$ -OO-a/-s,  $\Delta E_r$  of 296 297 anti-O<sub>2</sub>-addition is -5.8 kcal/mol, which is lower than that of syn-O<sub>2</sub>-addition by 0.4 298 kcal/mol, suggesting that anti-O<sub>2</sub>-addition is preferable over syn-O<sub>2</sub>-addition in energy. For the reactions  $P_{S4-add1'}+O_2 \rightarrow P_{S4-add1'}-OO-a/-s$  and  $P_{S4-add1''}+O_2 \rightarrow$ 299 300  $P_{S4-add1}$ "-OO-a/-s, it can be concluded the same by the values of  $\Delta E_r$  that 301 anti-O<sub>2</sub>-addition reaction is energetically feasible. The resulting P<sub>S4-add1</sub>-OO can proceed intramolecular cyclization reaction, where 302 303 the attack of end-site oxygen atom of the -OO group on C<sub>2</sub>-site of the C<sub>2</sub>=C<sub>3</sub> double bond, leading to the formation of peroxide bicyclic alkyl radicals.  $\Delta E_a$  and  $\Delta E_r$  of the 304 reaction  $P_{S4-add1}$ -OO-a  $\rightarrow P_{S4-add1}$ -OO-a-1 are 11.8 and -16.8 kcal/mol, respectively, 305 which are lower than those of the reaction  $P_{S4-add1}$ -OO-s  $\rightarrow P_{S4-add1}$ -OO-s-1 by 3.9 and 306 2.2 kcal/mol, respectively. The aforementioned results reveal that the intramolecular 307 cyclization reaction of anti-O<sub>2</sub>-addition product P<sub>S4-add1</sub>-OO-a is favorable on both 308

steric hindrance in the latter case. Among all the OH-addition reactions occurring at

309 thermochemically and kinetically. A similar conclusion is also derived from the intramolecular cyclization reactions of anti-O2-addition products PS4-add1'-OO-a and 310 P<sub>S4-add1</sub>"-OO-a. Notably, the barriers of the intramolecular cyclization reactions 311  $P_{S4-add1}$ -OO-a  $\rightarrow$   $P_{S4-add1}$ -OO-a-1 ( $\Delta E_a = 31.1$  kcal/mol) and  $P_{S4-add1}$ -OO-a  $\rightarrow$ 312  $P_{S4-addl}$ -OO-a-2 ( $\Delta E_a = 34.6$  kcal/mol) are extremely high, making them insignificant 313 in the atmosphere. The tautomerization between P<sub>S4-add1</sub>-OO-a-1 and P<sub>S4-add1</sub>"-OO-a-1 314 readily occurs due to the existence of resonance structures, and it is therefore that the 315 latter conformer is selected as a prototype for the investigating of its subsequent 316 reaction mechanism. 317 The formed PS4-add1"-OO-a-1 can combine with an O2 molecule leading to the 318 319 third generation peroxy radicals (also called as peroxide bicyclic peroxy radicals, BPR) 320 P<sub>S4-add1</sub>"-OO-a-2, and the lowest energy conformer is presented in Figure S6. The isomerization of P<sub>S4-add1"</sub>-OO-a-2 may undergo through a concerted process of the 321 322 cleavage of -O-O- bridge bond and C<sub>1</sub>-C<sub>2</sub> bond as well as hydrogen atom transfer 323 from the hydroxyl group to the bridge oxygen atom, yielding a new peroxy radical ( $\Delta E_{\rm a}$  = 28.5 kcal/mol). The room temperature rate coefficient is calculated to be 3.0  $\times$ 324 10<sup>-9</sup> s<sup>-1</sup>, which is several orders of magnitude low than the typical pseudo-first-order 325 rate constants  $k'_{\text{RO2+HO2}}$  (~ 0.01 s<sup>-1</sup>) and  $k'_{\text{RO2+NO}}$  (~ 0.01 s<sup>-1</sup>), suggesting that the 326 isomerization reaction is insignificance in the atmosphere. Therefore, the bimolecular 327 reactions of P<sub>S4-add1</sub>"-OO-a-2 are mainly taken into consideration in this study. 328 In the pristine environments, P<sub>S4-add1</sub>"-OO-a-2 can react with HO<sub>2</sub> radicals 329 resulting in the formation of the second generation products, bicyclic hydroperoxide 330 2<sup>nd</sup>-ROOH (S6) and peroxide bicyclic alkoxy radical (BAR) P<sub>S4-add1</sub>"-OO-a-3, as 331 depicted in Figure S6. For the subsequent reactions of S6 initiated by OH radicals, the 332 detailed mechanisms are discussed in Section 3.3.1. From Figure 3, it can be seen that 333 334 the unimolecular decomposition of P<sub>S4-add1</sub>"-OO-a-3 involves two kinds of pathways. One is the ring-opening reaction, where the breakage of  $C_5$ - $C_6$  bond produces an alkyl 335 radical S7 ( $\Delta E_a = 5.9$  kcal/mol<sub>1</sub>). The other is cyclization reaction, where the attack of 336 oxygen atom of O-centered site on the C<sub>4</sub>-site of the C<sub>3</sub>=C<sub>4</sub> double bond generates the 337 ring-retaining alkyl radical S15 ( $E_a = 8.0 \text{ kcal/mol}$ ). The branching ratios for the 338

339 formation of S7 and S15 are predicted to be 74.7% and 25.3%, respectively. As shown in Figure 3, S7 decomposes through the barrierless rupture of -O-O-340 bridge bond to form alkoxy radical S8-x, which includes five possible conformers as 341 342 presented in Figure S7. The Boltzmann populations of different conformers are listed in Table S2. S8-x can undergo intramolecular H-shifts, in which a hydrogen atom is 343 transferred from different carbon atoms to O-centered site, forming the alkyl radicals. 344 Among the competing H-shift reactions, 1,5 H-shift occurring at the -C<sub>5</sub>(O)H group 345 exhibits the smallest barrier ( $\Delta E_a = 0.6$  kcal/mol), and  $k_{\text{MC-TST}}$  is calculated to be 8.2 × 346 10<sup>9</sup> s<sup>-1</sup> at ambient temperature (Table S3). The formed S8-e-P can readily isomerize to 347 S9 due to its resonance stabilized structure, followed by decomposition into a 348 ketene-enol S10 and an alkyl radical S11 ( $\Delta E_a = 16.1 \text{ kcal/mol}$ ), or dissociation to S13 349 and CO ( $\Delta E_a = 29.4 \text{ kcal/mol}$ ). S11 can further react with O<sub>2</sub> leading to a HO<sub>2</sub> radical 350 and a hydroperoxide S12 that possessed a hydroxyl and two carbonyl groups ( $\Delta E_a$  = 351 352 14.0 kcal/mol). Alternatively, S8-x can proceed through the C<sub>1</sub>-C<sub>2</sub> bond scission to yield an unsaturated 1,4-dicarbonyl species S14 and an alkyl radical S11 ( $\Delta E_a = 2.2$ 353 kcal/mol), with the rate coefficient of  $2.1 \times 10^{10}$  s<sup>-1</sup>. Notably, both the 1,5 aldehyde 354 355 H-shift and C<sub>1</sub>-C<sub>2</sub> bond scission reactions yield a closed-shell species S12 with up to five oxygen atoms, and the branching ratios are predicted to be 28.1% and 71.9%, 356 respectively. The result is further supported by the previous study that the proportion 357 of aldehyde H-shift products constitutes about one third of the total products in the 358 reaction benzene + OH (Wang et al., 2020). 359 360 As shown in Figure 3, S15 can further react with O<sub>2</sub> leading to the fourth generation peroxy radical S16-x, which can proceed either intramolecular H-shifts 361 forming QOOH (Figure S8), or reactions with RO<sub>2</sub> radicals forming alkoxyl radical 362 S17. Notably, the barriers of intramolecular H-shifts are significantly high ( $\Delta E_a > 34.6$ 363 kcal/mol), making them less importance in the atmosphere. The transformation of S17 364 undergoes through the breakage of  $C_2$ - $C_3$  bond to produce S18 ( $\Delta E_a = 9.4$  kcal/mol), 365 followed by fragmentation into an alkoxyl radical S19 via the barrierless rupture of 366 the -O-O- bridge bond. Then, S19 dissociates to an OH radical, a glycolaldehyde S25 367 and a C<sub>6</sub>-epoxide product S23 bearing a hydroxy and three carbonyl groups, being the 368

dominant pathway. The regeneration of OH radicals drives the successive autoxidation of styrene, eventually leading to the formation of the highly oxidized products.

**Figure 2.** PES for the oxidation of  $1^{st}$ -ROOH(S4) initiated by OH radicals at the M06-2X/6-311++G(3df,3pd)//M06-2X/6-31+g(d,p) level

 $\label{eq:Figure 3.PES} \textbf{Figure 3.} \ PES \ for the unimolecular decomposition of } P_{S4-add1''}\text{-OO-a-3} \ and its subsequent reactions at the } M06-2X/6-311++G(3df,3pd)//M06-2X/6-31+g(d,p) \ level$ 

## 3.2.2 The oxidation mechanism of 1st-RONO2 initiated by OH

## radicals

372

373374

381

The OH-initiated oxidation of  $1^{st}$ -RONO<sub>2</sub> (S5) proceeds through the addition of OH radicals to different carbon sites in the benzene ring to form various alkyl radicals  $P_{S5-addx}$ , as depicted in Figure 4. Among the competing OH-addition reactions, the OH-addition reaction at the C1-site, which proceeds in the opposite direction as the – ONO<sub>2</sub> group, has the smallest barrier ( $R_{S5-add1}$ ,  $\Delta E_a = 0.4$  kcal/mol) due to the stability

413

414

reaction.

of the formed product P<sub>S5-add1</sub>. The result again shows that the *ortho*-addition reaction 386 is energetically feasible. P<sub>S5-add1</sub> may isomerize to two other resonance structures, namely, P<sub>S5-add1</sub> and P<sub>S5-add1</sub>. For the reaction P<sub>S5-add1</sub> + O<sub>2</sub>, O<sub>2</sub> may add on either the 387 388 opposite (anti-O<sub>2</sub>-addition) or the same direction (syn-O<sub>2</sub>-addition) relative to the – NO<sub>3</sub> group, leading to the second generation peroxyl radicals P<sub>S5-add1</sub>-OO-a and 389 P<sub>S5-add1</sub>-OO-s (Figure S9). The exoergicity of these two reactions are 6.7 and 4.4 390 kcal/mol, respectively, suggesting that the anti-O2-addition reaction is 391 thermochemically favorable. Next, they can isomerize via a cyclization process to 392 yield  $P_{S5-add1}$ -OO-a-1 and  $P_{S5-add1}$ -OO-s-1 with the  $\Delta E_a$  of 13.3 and 18.1 kcal/mol. This 393 394 result shows that the cyclization reaction of anti-O<sub>2</sub>-addition product P<sub>S5-add1</sub>-OO-a is kinetically feasible. A similar conclusion is also obtained from the reaction P<sub>S5-add1"</sub> + 395 396 O<sub>2</sub> that the formation of anti-O<sub>2</sub>-addition product P<sub>S5-add1"</sub>-OO-a-1 is dominant. Due to 397 the existence of the conjugate double bond, it facilitates the tautomerization between 398 P<sub>S5-add1</sub>-OO-a-1 and P<sub>S5-add1</sub>"-OO-a-1. Therefore, we mainly focus on the subsequent 399 chemistry of P<sub>S5-add1</sub>"-OO-a-1 in the present study. P<sub>S5-add1</sub>"-OO-a-1 can further react with an O<sub>2</sub> molecule leading to the third 400 401 generation peroxyl radicals P<sub>S5-add1</sub>"-OO-a-2, which include multiple conformers. The lowest energy conformer resulting from conformer search is presented in Figure S10. 402 In urban environments, the bimolecular reaction of PS5-add1"-OO-a-2 with NO yields 403 the second generation products, bicyclic organic nitrate 2<sup>nd</sup>-RONO<sub>2</sub> (S26) and BAR 404 Ps5-add1"-OO-a-3 as displayed in Figure S10. The detailed mechanism of OH-initiated 405 oxidation of S26 is discussed in Section 3.3.2. As shown in Figure 5, Ps5-add1"-OO-a-3 406 407 can either proceed via a ring opening process to form an alkyl radical S27 ( $\Delta E_a = 7.3$ kcal/mol), or undergo through a cyclization process to generate an epoxide species 408 S35 ( $\Delta E_a = 8.5 \text{ kcal/mol}$ ). The branching ratios of these two reactions are predicted to 409 410 be 69.2% and 30.8%, respectively. Notably, the branching ratio of cyclization reaction of P<sub>S5-add1</sub>"-OO-a-3 increases by 5.5% compared to that of cyclization reaction of 411

P<sub>S4-add1</sub>"-OO-a-3, suggesting that the -ONO<sub>2</sub> substitution is beneficial to cyclization

415 bond to form S28-x, and the Boltzmann populations of different conformers are listed in Table S4. S28-x can undergo via various intramolecular H-shifts to produce QOOH, 416 in which hydrogen atom transfer from the -C(O)H group to the terminal oxygen atom 417 418 has the smallest barrier ( $\Delta E_a = 2.0$  kcal/mol). Alternatively, S28-x can proceed through the cleavage of C<sub>1</sub>-C<sub>2</sub> bond to generate an unsaturated 1,4-dicarbonyl 419 compound S34 and an alkyl radical S31 (Figure S11). The rare coefficients of these 420 two pathways are predicted to be  $1.7 \times 10^9$  and  $5.8 \times 10^9$  s<sup>-1</sup> (Table S5), respectively, 421 with the branching ratios of 23% and 77%. The formed S31 may undergo through 422 H-abstraction by O<sub>2</sub> to yield an organic nitrate S32 bearing a hydroxyl and two 423 carbonyl groups ( $\Delta E_a = 11.7 \text{ kcal/mol}$ ). 424 S35 can combine with an O<sub>2</sub> molecule forming the fourth generation peroxyl 425 radicals S36-x, which have five possible conformers. S36-x can proceed either 426 intramolecular H-shifts forming QOOH, or reaction with RO2 radicals generating 427 428 alkoxyl radical S37. However, the barriers of intramolecular H-shifts are extremely 429 high ( $\Delta E_a > 31.3$  kcal/mol), making them less importance in the atmosphere (Figure S12). The degradation of S37 initially proceeds via the breakage of C2-C3 bond to 430 431 form S38 ( $\Delta E_a = 9.2$  kcal/mol), followed by decomposition into an alkoxyl radical S39 via the barrierless scission of -O-O- bridge bond. The dominant pathway of the 432 433 unimolecular decomposition of S39 is the formation of a glyoxal and a C6-epoxide species S40-1 bearing a -NO<sub>3</sub>, a hydroxyl and two carbonyl groups. This process 434 differs from the unimolecular decay of S19, where the favorable pathways is the 435 formation of a tricarbonyl compound S23. The aforementioned results reveal that the 436 437 preferable pathway is strongly dependent on the breakage of C-C bond associated with the property of substituents in the decomposition of alkoxy radicals. 438

441

442

443 444

**Figure 4.** PES for the oxidation of  $1^{st}$ -RONO<sub>2</sub>(S5) initiated by OH radicals at the M06-2X/6-311++G(3df,3pd)//M06-2X/6-31+g(d,p) level

Figure 5. PES for the unimolecular decomposition of  $P_{S5-add1}$ —OO-a-3 and its subsequent reactions at the M06-2X/6-311++G(3df,3pd)/M06-2X/6-31+g(d,p) level

# 3.3 Third generation OH oxidation mechanisms of 2<sup>nd</sup>-ROOH

## and 2<sup>nd</sup>-RONO<sub>2</sub>

The second generation products, bicyclic hydroperoxide 2<sup>nd</sup>-ROOH and bicyclic organic nitrate 2<sup>nd</sup>-RONO<sub>2</sub>, have multiple conformers. The global minimum structures of 2<sup>nd</sup>-ROOH (S6) and 2<sup>nd</sup>-RONO<sub>2</sub> (S26) resulting from the conformer search are presented in Figures S6 and S10, respectively.

## 3.3.1 The oxidation mechanism of 2<sup>nd</sup>-ROOH initiated by OH

#### radicals

455

459

470

472

474

480

OH-initiated oxidation of 2<sup>nd</sup>-ROOH (S6) undergoes through the addition of OH radicals to either side of the  $C_3=C_4$  double bond to generate the alkyl radicals. In this context, syn-OH-addition is defined as the scenario in which the addition of OH radicals occurs at the same side as the -OOH group, while anti-OH-addition is referred to the scenario in which the addition of OH radicals occurs at the opposite side as the -OOH group. As shown in Figure S13, the syn-OH-addition occurring at the C<sub>3</sub>-site in the C<sub>3</sub>=C<sub>4</sub> double bond has the smallest barrier ( $\Delta E_a = 2.4$  kcal/mol). Next, the unimolecular decay of syn-OH-addition product P<sub>S6-add3</sub> proceeds through a cyclization process to yield an epoxide compound S41 and an OH radical byproduct  $(\Delta E_{\rm a} = 15.3 \text{ kcal/mol})$ , or undergoes via intramolecular 1,4 H-shift to form a peroxy radical S42 ( $\Delta E_a = 21.8 \text{ kcal/mol}$ ), or proceeds via the elimination of hydrogen atom to produce an alkene S43 ( $\Delta E_a = 37.9 \text{ kcal/mol}$ ), as depicted in Figure S14. Based on the values of  $\Delta E_a$ , the dominant pathway of the unimolecular decomposition of  $P_{S6-add3}$ is the formation of S41, and the rate coefficient  $k_{R41}$  is calculated to be 1.8  $\times$  10<sup>2</sup> s<sup>-1</sup>. The pseudo-first-order rate constant  $k'_{R+O2}$  of the bimolecular reactions of typical alkyl radicals with  $O_2$  is  $3.0 \times 10^7$  s<sup>-1</sup> (Ma et al., 2021), which is about five orders of magnitude greater than  $k_{R41}$ , suggesting that the unimolecular decomposition of P<sub>S6-add3</sub> is insignificant. As shown in Figure 6, the fourth generation peroxy radicals S44-x formed in the addition reaction P<sub>S6-add3</sub> + O<sub>2</sub> can either proceed via intramolecular H-shits to form QOOH, or undergo through self- or cross-reactions to yield an alkoxy radical S45. Due to the considerably high barriers of intramolecular H-shifts, they are deemed to be negligible under atmospheric conditions. S45 can convert into an alkyl radical S46 through the cleavage of C<sub>4</sub>-C<sub>5</sub> bond, or dissociate to an alkyl radical S50 via the rupture of C<sub>3</sub>-C<sub>4</sub> bond. The barrier of the former reaction is 3.9 kcal/mol, which is lower than that of the latter pathway by 2.6 kcal/mol, indicating that the formation of S46 is kinetically preferable. Then, S46 decomposes into an OH radical byproduct and a C<sub>8</sub>-product S47 bearing a -OOH, a peroxide bridge, two carbonyls, and three

hydroxy groups, which is expected to be the dominant pathway owing to its lower barrier. The rate coefficient  $k_{RS47}$  is estimated to be  $1.8 \times 10^9$  s<sup>-1</sup>, which is about two orders of magnitude greater than the pseudo-first-order rate constant  $k'_{R+O2}$  ( $3.0 \times 10^7$  s<sup>-1</sup>). The result reveals that the unimolecular decomposition of S46 is more competitive than the bimolecular reaction with  $O_2$ . The formed OH radicals can once again participate in the oxidations of styrene and its highly oxidized products, continuing these processes until they are completely consumed.

$$\begin{array}{c} \text{OH} \quad \text{OOH} \\ \text{Id-Hohiff} \\ \text{HO} \quad \text{OOH} \\ \text{S44-g.} \quad \text{OH} \\ \text{S45-g.} \quad \text{OOH} \\ \text{S46-g.} \quad \text{OOH} \\ \text{OOH} \\$$

**Figure 6.** PES for the subsequent reactions of  $P_{S6-add3}$  in the presence of  $O_2$  at the M06-2X/6-311++G(3df,3pd)//M06-2X/6-31+g(d,p) level

# 3.3.2 The oxidation mechanism of 2<sup>nd</sup>-RONO<sub>2</sub> initiated by OH

## radicals

A schematic PES for the OH-initiated oxidation of  $2^{nd}$ -RONO $_2$  (S26) is presented in Figure S15. The bimolecular reaction of S26 with OH radicals encompasses four possible routes, namely,  $R_{S26\text{-add3}}$ ,  $R_{S26\text{-add4}}$ ,  $R_{S26\text{-add3}}$  and  $R_{S26\text{-add4}}$ . Among these four OH-addition reactions,  $R_{S26\text{-add3}}$  has the smallest barrier ( $\Delta E_a = 2.4$  kcal/mol), which is in line with the reaction S6 + OH that syn-OH-addition reaction occurring at the

498 C<sub>3</sub>-site is predominant. The syn-OH-addition product P<sub>S26-add3</sub> has three potential unimolecular decay pathways, as depicted in Figure S16: (1) P<sub>S26-add3</sub> dissociates to an 499 epoxide S52 and a NO<sub>2</sub> molecule through a cyclization process ( $\Delta E_a = 18.5 \text{ kcal/mol}$ ); 500 501 (2)  $P_{S26-add3}$  isomerizes to an alkyl radical S53 via the intramolecular 1,2 H-shift ( $\Delta E_a$ = 40.0 kcal/mol); (3) P<sub>S26-add3</sub> converts into an alkene S54 via the elimination of 502 hydrogen atom ( $\Delta E_a = 39.1$  kcal/mol). Based on the values of  $\Delta E_a$ , the dominant 503 pathway of the unimolecular decomposition of P<sub>S26-add3</sub> is the formation of S52. The 504 rate coefficient  $k_{R52}$  is calculated to be 0.4 s<sup>-1</sup>, which is about seven orders of 505 magnitude lower than the pseudo-first-order rate constant  $k'_{R+O2}$ , indicating that the 506 unimolecular decomposition of P<sub>S26-add3</sub> is less importance. 507 In the presence of O2, PS26-add3 can react with an O2 molecule leading to the 508 509 formation of the fourth generation peroxy radicals S55-x, which has seven possible conformers, as shown in Figure 7. For the intramolecular H-shifts of S55-x, not all of 510 511 reactants (S55-c, S55-d and S55-e) have the suitable conformers that allow for the 512 pathways across the reaction barriers. The barriers of intramolecular H-shifts are considerably high ( $\Delta E_a = 20.8$  kcal/mol), making them uncompetitive in the 513 514 atmosphere. Alternatively, S55-x can react with other RO<sub>2</sub> radicals forming an alkoxyl radical S56, followed by decomposition into an alkyl radical S57 via the breakage of 515  $C_4$ - $C_5$  bond ( $\Delta E_a = 2.0$  kcal/mol), or fragmentation into an alkyl radical S61 through 516 the cleavage of  $C_3$ - $C_4$  bond ( $\Delta E_a = 4.5$  kcal/mol). The aforementioned results reveal 517 that the formation of S57 is energetically favorable, which is consistent with the 518 519 conclusion derived from the unimolecular decomposition of S45 that the breakage of 520 C<sub>4</sub>-C<sub>5</sub> bond is feasible. Next, S57 dissociates to a NO<sub>2</sub> coproduct and a C<sub>8</sub>-product S58 that possessed a -NO<sub>3</sub>, a peroxide bridge, two carbonyls, and three hydroxy 521 groups. This pathway is expected to be the dominant one ( $\Delta E_a = 1.5 \text{ kcal/mol}$ ), with 522 the rate coefficient  $k_{RS58}$  of  $1.2 \times 10^9$  s<sup>-1</sup>. The resulting NO<sub>2</sub> can further participate in 523 the cycling of NO<sub>x</sub>, ultimately generating tropospheric ozone and SOA. 524

$$\begin{array}{c} 1.4 + 1.4 \sin t \\ 1.0 & 0.00 \\ 8.5 + 3 + 1.5 \\ 1.5 + 1.4 \sin t \\ 1.5 + 1.5 \sin t \\ 1.5 + 1.4 \sin t \\ 1.5 + 1.5 \sin t \\ 1.5 + 1.5$$

**Figure 7.** PES for the subsequent reactions of  $P_{S26-add3}$  in the presence of  $O_2$  at the M06-2X/6-311++G(3df,3pd)//M06-2X/6-31+g(d,p) level

## 3.4 Volatility

The saturated vapour pressure ( $P^0$ ) and saturated concentration ( $c^0$ ) of styrene and its multi-generation OH oxidation closed-shell products are predicted by using the SIMPOL.1 method (Pankow et al., 2008), and the results are listed in Table S6. The  $P^0$  and  $C^0$  of styrene are estimated to be  $4.63 \times 10^{-3}$  atm and  $1.95 \times 10^7$  ug/m³, respectively, which are 4-5 orders of magnitude greater than those of the first generation products S4 and S5. Based on the Volatility Basis Set (VBS) of organic compounds as proposed by Donahue et al., (2012) S4 and S5 are classified as intermediate volatility organic compounds (IVOCs), which exist exclusively in the gas phase under atmospheric conditions (Bianchi et al., 2019). The second generation products mainly include the ring-opening (S10, S12, S14, S20, S23, S32, and S40-1) and ring-retaining species (S6 and S26). Based on the values of  $c^0$ , S12, S23 and S32 belong to IVOCs, while S20 and S40-1 belong to semivolatile organic compounds

(SVOC). The difference in volatility is attributed to the fact that the carbon number and functional group of the former case are significantly lower than those of the latter case. S6 and S26 are classified as low volatility organic compounds (LVOCs), which can condense onto the existing large particles (Bianchi et al., 2019). Notably, the third generation products S47 and S51 are also classified as LVOCs, but their  $c^0$  values are two orders of magnitude lower than those of the second generation product S6. A similar conclusion is also obtained when comparing the  $c^0$  values of the third generation products S58 and S62 with those of the second generation product S26. The aforementioned results reveal that the volatility of the multiple generation OH oxidation products significantly decreases with increasing the number of OH oxidation steps. As the oxidation reactions of the third generation products proceed further, the formed products may possess sufficiently low volatility to participate in the formation and growth of new aerosol particle.

#### 4 Conclusions

The formation mechanisms of highly oxidized products from the multi-generation OH oxidation of styrene under different  $NO_x$  conditions are investigated by mean of quantum chemical methods. The primary conclusions are summarized as follows.

(a) For the first generation OH oxidation of styrene, the addition of OH radicals to terminal carbon ( $C_{\beta}$ -site) of a vinyl group is the dominant pathway. For the isomerization of the first generation RO<sub>2</sub> radicals formed in the association reaction of OH-addition product with O<sub>2</sub>, it can proceed through intramolecular hydrogen atom transfer from the –OH group to the terminal oxygen atom to form a hydroperoxyalkoxy radical, followed by decomposition into benzaldehyde by the successive elimination of HCHO and OH radicals. The bimolecular reactions of the first generation RO<sub>2</sub> radicals with HO<sub>2</sub> radicals and NO lead to the formation of the first generation closed-shell products 1<sup>st</sup>-ROOH ( $C_8H_{10}O_3$ ), benzaldehyde ( $C_7H_6O$ ), and 1<sup>st</sup>-RONO<sub>2</sub> ( $C_8H_9NO_3$ ).

569 (b) For the second generation OH oxidation of 1<sup>st</sup>-ROOH, OH-addition occurring at the opposite direction of -OOH group exhibit significantly lower barriers 570 compared to those at the same direction of -OOH group. The addition of OH radicals 571 to C<sub>1</sub>-site has the smallest barrier among the competing OH-addition reactions. BPR 572 formed by two O<sub>2</sub>-addition steps and a cyclization process can either react with 573 RO2 · and NO to produce BAR, or interact with HO2 · and NO to form the second 574 generation closed-shell products 2<sup>nd</sup>-ROOH (C<sub>8</sub>H<sub>12</sub>O<sub>8</sub>) and 2<sup>nd</sup>-RONO<sub>2</sub> (C<sub>8</sub>H<sub>10</sub>N<sub>2</sub>O<sub>10</sub>). 575 The formed BAR may undergo via the ring-opening and subsequent decomposition 576 reactions to generate an unsaturated 1,4-dicarbonyl compound C<sub>4</sub>H<sub>4</sub>O<sub>2</sub>, a ketene-enol 577 C<sub>4</sub>H<sub>4</sub>O<sub>2</sub>, and a hydroperoxide C<sub>4</sub>H<sub>6</sub>O<sub>5</sub> bearing a hydroxyl and two carbonyl groups. 578 579 Alternatively, it can proceed through the cyclization and subsequent dissociation reactions to produce a glycolaldehyde and a epoxide C<sub>6</sub>H<sub>6</sub>O<sub>5</sub> containing a hydroxy 580 and three carbonyl groups. The branching ratios of these two pathways are 74.7% and 581 582 25.3%, respectively. (c) For the second generation OH oxidation of 1<sup>st</sup>-RONO<sub>2</sub>, the OH-addition 583 reaction at the C1-site, which proceeds in the opposite direction as the -ONO2 group, 584 585 has the smallest barrier. BAR formed in the reactions of BPR with HO2 radicals and NO can proceed either the ring opening or cyclization process to from alkyl radicals 586 587 S27 and S35, with the branching ratio of 69.2% and 30.8%. The main products of the decomposition of S27 are an unsaturated 1,4-dicarbonyl compound C<sub>4</sub>H<sub>4</sub>O<sub>2</sub> and an 588 organic nitrate C<sub>4</sub>H<sub>5</sub>NO<sub>6</sub> bearing a hydroxyl and two carbonyl groups. The primary 589 products of the dissociation of S35 are a glyoxal and a C<sub>6</sub>-epoxide specie C<sub>6</sub>H<sub>7</sub>NO<sub>7</sub> 590 591 containing a –NO<sub>3</sub>, a hydroxyl and two carbonyl groups. (d) For the third generation OH oxidation of 2<sup>nd</sup>-ROOH and 2<sup>nd</sup>-RONO<sub>2</sub>, 592 syn-OH-addition reactions occurring at the C<sub>3</sub>-site are predominant. The alkoxyl 593 radical formed in the reaction 2<sup>nd</sup>-ROOH + OH decomposes into an OH radical 594 byproduct and a C<sub>8</sub>-product, C<sub>8</sub>H<sub>12</sub>O<sub>9</sub>, bearing a -OOH, a peroxide bridge, two 595 carbonyls, and three hydroxy groups. The alkoxyl radical formed in the reaction 596 2<sup>nd</sup>-RONO<sub>2</sub> + OH dissociates to a NO<sub>2</sub> co-product and a C<sub>8</sub>-product, C<sub>8</sub>H<sub>11</sub>NO<sub>10</sub>, 597 containing a -NO<sub>3</sub>, a peroxide bridge, two carbonyls, and three hydroxy groups. 598

(e) The first generation products  $1^{st}$ -ROOH and  $1^{st}$ -RONO<sub>2</sub> belong to IVOCs, which exist exclusively in the gas phase. In the numerous second generation products, only  $2^{nd}$ -ROOH and  $2^{nd}$ -RONO<sub>2</sub> are identified as IVOCs. Although the third generation products are also classified as LVOC, their  $c^0$  values are several orders of magnitude lower than those of the second generation products. As a result, the volatility of the multiple generation OH oxidation products significantly decreases with increasing the number of OH oxidation steps.

## Data availability

The data are accessible by contacting the corresponding author (huangyu@ieecas.cn).

## Supplement

Tables S1, S2 and S4 list the relative electronic energy, free energy and Boltzmann population of different conformers involved in S2-1-x, S8-x and S28-x. Tables S3 and S5 list the MC-TST rate coefficients for the intramolecular H-shift reactions of S8-x and S28-x. Table S6 summaries the saturated vapour pressure and saturated concentrations of styrene and its multiple generation OH oxidation closed-shell products. Figures S1-S3 display the PESs for the unimolecular reactions of S2-2-x, S2-3-x and S2-4-x. Figure S4 shows the global minimum structures of 1<sup>st</sup>-ROOH(S4) and 1<sup>st</sup>-RONO<sub>2</sub>(S5). Figures S5 and S9 show the PESs for the addition reactions P<sub>S4-add1</sub> + O<sub>2</sub> and P<sub>S5-add1</sub> + O<sub>2</sub>, and their subsequent intramolecular cyclization reactions. Figures S6 and S10 present the lowest energy conformers of third generation peroxy radical P<sub>S4-add1</sub> "-OO-a-2 and P<sub>S5-add1</sub>"-OO-a-2. Figures S7-S8 depict the PESs for the intramolecular hydrogen transfer reactions of S8-x and S16-x. Figures S11 and S12 display the PESs for the intramolecular hydrogen transfer reactions of S28-x and S36-x. Figures S13 and S14 show the PESs for the OH-initiated oxidation of 2<sup>nd</sup>-ROOH (S6) and unimolecular decomposition of P<sub>S6-add3</sub>.

660

Figures S15 and S16 show the PESs for the OH-initiated oxidation of 2<sup>nd</sup>-RONO<sub>2</sub> 627 (S26) and unimolecular decomposition of P<sub>S26-add3</sub>. 628 629 **Author contribution** 630 LC and YH conceptualized the study. LC conducted quantum chemical 631 calculation. YX and ZJ analyzed the data. LC conducted the volatility estimation. All 632 authors discussed the results and commented on the manuscript. 633 634 **Competing interests** 635 The contact author has declared that none of the authors has any competing interests. 636 637 **Financial support** 638 639 This study was supported by the National Natural Science Foundation of China (grant nos. 42175134) and the Youth Innovation Promotion Association of the Chinese 640 641 Academy of Sciences (grant nos. 2022415). 642 Reference 643 Alecu, I. M., Zheng, J., Zhao, Y., and Truhlar, D. G.: Computational thermochemistry: scale factor 644 databases and scale factors for vibrational frequencies obtained from electronic model 645 646 chemistries, J. Chem. Theory Comput., 6, 2872-2887, https://doi.org/10.1021/ct100326h, 647 2010. 648 Bianchi, F., Kurt én, T., Riva, M., Mohr, C., Rissanen, M. P., Roldin, P., Berndt, T., Crounse, J. D., 649 Wennberg, P. O., Mentel, T. F., Wildt, J., Junninen, H., Jokinen, T., Kulmala, M., Worsnop, D. 650 R., Thornton, J. A., Donahue, N., Kjaergaard, H. G., and Ehn, M.: Highly oxygenated organic 651 molecules (HOM) from gas-phase autoxidation involving peroxy radicals: a key contributor 119, 652 atmospheric aerosol, Chem. Rev.. 3472-3509, to 653 https://doi.org/10.1021/acs.chemrev.8b00395, 2019. Bloss, C., Wagner, V., Jenkin, M. E., Volkamer, R., Bloss, W. J., Lee, J. D., Heard, D. E., Wirtz, K., 654 655 Martin-Reviejo, M., Rea, G., Wenger, J. C., and Pilling, M. J.: Development of a detailed chemical mechanism (MCMv3.1) for the atmospheric oxidation of aromatic hydrocarbons, 656 657 Atmos. Chem. Phys., 5, 641-664, https://doi.org/10.5194/acp-5-641-2005, 2005. 658 Cabrera-Perez, D., Taraborrelli, D., Sander, R., and Pozzer, A.: Global atmospheric budget of

https://doi.org/10.5194/acp-16-6931-2016, 2016.

simple monocyclic aromatic compounds, Atmos. Chem. Phys., 16, 6931-6947,

- Canneaux, S., Bohr, F., and Henon, E.: KiSThelP: a program to predict thermodynamic properties 662 and rate constants from quantum chemistry results, J. Comput. Chem., 35, 82-93, https://doi.org/10.1002/jcc.23470, 2013. 663
- Chen, L., Huang, Y., Xue, Y., Jia, Z., and Wang, W.: Atmospheric oxidation of 1-butene initiated 664 665 by OH radical: Implications for ozone and nitrous acid formations, Atmos. Environ., 244, 118010-118021, https://doi.org/10.1016/j.atmosenv.2020.118010, 2021. 666
- Cho, J., Roueintan, M., and Li, Z.: Kinetic and dynamic investigations of OH reaction with styrene, J. Phys. Chem. A, 118, 9460-9470, https://doi.org/10.1021/jp501380j, 2014. 668
- Donahue, N. M., Kroll, J. H., Pandis, S. N., and Robinson, A. L.: A two-dimensional volatility 669 670 basis set - Part 2: Diagnostics of organic-aerosol evolution, Atmos. Chem. Phys., 12, 615-634, https://doi.org/10.5194/acp-12-615-2012, 2012. 671
- Eckart, C.: The penetration of a potential barrier by electrons, Phys. Rev., 35, 1303-1309, 673 https://doi.org/10.1103/PhysRev.35.1303, 1930.
- Environmental Protection Agency (EPA). Clean Air Act: Title I-Air Pollution Prevention and 674 675 Control. U.S. 1990.
- Fern ández-Ramos, A., Ellingson, B. A., Meana-Pa ñeda, R., Marques, J. M. C., and Truhlar, D. G.: 676 Symmetry numbers and chemical reaction rates, Theor. Chem. Acc., 118, 813-826, 677 678 https://doi.org/10.1007/s00214-007-0328-0, 2007.
- Forstner, H. J. L., Flagan, R. C., and Seinfeld, J. H.: Secondary organic aerosol from the photooxidation of aromatic hydrocarbons: molecular composition, Environ. Sci. Technol., 31, 680 681 1345-1358, https://doi.org/10.1021/es9605376, 1997.
- Frisch, M. J., Trucks, G. W., Schlegel, H. B., Scuseria, G. E., Robb, M. A., Cheeseman, J. R., 683 Scalmani, G., Barone, V., Petersson, G. A., Nakatsuji, H., Li, X., Caricato, M., Marenich, A.
  - V., Bloino, J., Janesko, B. G., Gomperts, R., Mennucci, B., Hratchian, H. P., Ortiz, J. V.,
- Izmaylov, A. F., Sonnenberg, J. L., Williams-Young, D., Ding, F., Lipparini, F., Egidi, F., 685
- Goings, J., Peng, B., Petrone, A., Henderson, T., Ranasinghe, D., Zakrzewski, V. G., Gao, J., 686
- Rega, N., Zheng, G., Liang, W., Hada, M., Ehara, M., Toyota, K., Fukuda, R., Hasegawa, J.,
- Ishida, M., Nakajima, T., Honda, Y., Kitao, O., Nakai, H., Vreven, T., Throssell, K., 688
- Montgomery, J. A., Peralta, J. J. E., Ogliaro, F., Bearpark, M. J., Heyd, J. J., Brothers, E. N.,
- Kudin, K. N., Staroverov, V. N., Keith, T. A., Kobayashi, R., Normand, J., Raghavachari, K.,
- Rendell, A. P., Burant, J. C., Iyengar, S. S., Tomasi, J., Cossi, M., Millam, J. M., Klene, M.,
- Adamo, C., Cammi, R., Ochterski, J. W., Martin, R. L., Morokuma, K., Farkas, O., Foresman, 693
- J. B., and Fox, D. J.: Gaussian 16, Revision B.01, Gaussian, Inc., Wallingford CT, 2016.
- Fu, Z., Guo, S., Xie, H. B., Zhou, P., Boy, M., Yao, M., and Hu, M.: A near-explicit reaction 695 mechanism of chlorine-initiated limonene: implications for health risks associated with the concurrent use of cleaning agents and disinfectants, Environ. Sci. Technol., 58, 19762-19773, 696
- https://doi.org/10.1021/acs.est.4c04388, 2024.
- Fu, Z., Ma, F., Liu, Y., Yan, C., Huang, D., Chen, J., Elm, J., Li, Y., Ding, A., Pichelstorfer, L., Xie, 699 H. B., Nie, W., Francisco, J. S., and Zhou, P.: An overlooked oxidation mechanism of toluene: 700 computational predictions and experimental validations, Chem. Sci., 14, 13050-13059, 701 https://doi.org/10.1039/D3SC03638C, 2023.
- Fu, Z., Xie, H. B., Elm, J., Guo, X., Fu, Z., and Chen, J.: Formation of low-volatile products and 703 unexpected high formaldehyde yield from the atmospheric oxidation of methylsiloxanes,
- Environ. Sci. Technol., 54, 7136-7145, https://doi.org/10.1021/acs.est.0c01090, 2020.

- Fukui, K.: The path of chemical reactions the IRC approach, Acc. Chem. Res., 14, 363-368,
   https://doi.org/10.1021/ar00072a001, 1981.
- Garmash, O., Rissanen, M. P., Pullinen, I., Schmitt, S., Kausiala, O., Tillmann, R., Zhao, D.,
- Percival, C., Bannan, T. J., Priestley, M., Hallquist, Å M., Kleist, E., Kiendler-Scharr, A.,
- Hallquist, M., Berndt, T., McFiggans, G., Wildt, J., Mentel, T. F., and Ehn, M.:
- Multi-generation OH oxidation as a source for highly oxygenated organic molecules from
- aromatics, Atmos. Chem. Phys., 20, 515-537, https://doi.org/10.5194/acp-20-515-2020, 2020.
- Gilbert, R. G., and Smith, S. C.: Theory of unimolecular and recombination reactions, Blackwell
   Scientific: Carlton, Australia, 1990.
- Glowacki, D. R., Liang, C. H., Morley, C., Pilling, M. J., and Robertson, S. H.: MESMER: an
- open-source master equation solver for multi-energy well reactions, J. Phys. Chem. A, 116,
- 9545-9560, https://doi.org/10.1021/jp3051033, 2012.
- Holbrook, K. A., Pilling, M. J., Robertson, S. H., and Robinson, P. J.: Unimolecular reactions, 2nd
   ed.; Wiley: New York, 1996.
- Huang, Y., Su, T., Wang, L., Wang, N., Xue, Y., Dai, W., Lee, S. C., Cao, J., and Ho, S. S. H.:
- Evaluation and characterization of volatile air toxics indoors in a heavy polluted city of
- northwestern China in wintertime, Sci. Total Environ., 662, 470-480 722 https://doi.org/10.1016/j.scitotenv.2019.01.250, 2019.
- Hyttinen, N., Kupiainen-Määtä, O., Rissanen, M. P., Muuronen, M., Ehn, M., and Kurtén, T.:
- Modeling the charging of highly oxidized cyclohexene ozonolysis products using
- nitrate-based chemical ionization, J. Phys. Chem. A, 119, 6339-6345,
- https://doi.org/10.1021/acs.jpca.5b01818, 2015.
- Iuga, C., Galano, A., and Vivier-Bunge, A.: Theoretical investigation of the OH-initiated oxidation
- of benzaldehyde in the troposphere, Chem. Phys. Chem., 9, 1453-1459,
- https://doi.org/10.1002/cphc.200800144, 2008.
- Iyer, S., Kumar, A., Savolainen, A., Barua, S., Daub, C., Pichelstorfer, L., Roldin, P., Garmash, O.,
- Seal, P., Kurt én, T., and Rissanen, M.: Molecular rearrangement of bicyclic peroxy radicals is
- a key route to aerosol from aromatics, Nat. Commun., 14, 4984-4991,
- https://doi.org/10.1038/s41467-023-40675-2, 2023.
- Ji, Y., Zhao, J., Terazono, H., Misawa, K., Levitt, N. P., Li, Y., Lin, Y., Peng, J., Wang, Y., Duan, L.,
- Pan, B., Zhang, F., Feng, X., An, T., Marrero-Ortiz, W., Secrest, J., Zhang, A. L., Shibuya, K.,
- Molina, M. J., and Zhang, R.: Reassessing the atmospheric oxidation mechanism of toluene,
- Proc. Natl. Acad. Sci. U.S.A., 114, 8169-8174, https://doi.org/10.1073/pnas.1705463114,
   2017.
- Ji, Y., Zheng, J., Qin, D., Li, Y., Gao, Y., Yao, M., Chen, X., Li, G., An, T., and Zhang, R.:
- OH-initiated oxidation of acetylacetone: implications for ozone and secondary organic
- aerosol formation, Environ. Sci. Technol., 52, 11169-11177,
- https://doi.org/10.1021/acs.est.8b03972, 2018.
- Koppmann, R.: Volatile organic compounds in the atmosphere, John Wiley & Sons, 2008.
- Li, M., Zhang, Q., Zheng, B., Tong, D., Lei, Y., Liu, F., Hong, C., Kang, S., Yan, L., Zhang, Y., Bo,
- Y., Su, H., Cheng, Y., and He, K.: Persistent growth of anthropogenic non-methane volatile
- organic compound (NMVOC) emissions in China during 1990-2017: drivers, speciation and
- ozone formation potential, Atmos. Chem. Phys., 19, 8897-8913,
- https://doi.org/10.5194/acp-19-8897-2019, 2019.

- Lu, T.: Molclus program, Version 1.9.3. http://www.keinsci.com/research/molclus.html (accessed
   May 21, 2024).
- 751 Ma, F., Guo, X., Xia, D., Xie, H. B., Wang, Y., Elm, J., Chen, J., and Niu, J.: Atmospheric 752 chemistry of allylic radicals from isoprene: a successive cyclization-driven autoxidation
- mechanism, Environ. Sci. Technol., 55, 4399-4409, https://doi.org/10.1021/acs.est.0c07925, 2021.
- Møler, K. H., Berndt, T., and Kjaergaard, H. G.: Atmospheric autoxidation of amines, Environ.
   Sci. Technol., 54, 11087-11099, https://doi.org/10.1021/acs.est.0c03937, 2020.
- Møller, K. H., Otkjær, R. V., Hyttinen, N., Kurtén, T., and Kjaergaard, H. G.: Cost-effective implementation of multiconformer transition state theory for peroxy radical hydrogen shift reactions, J. Phys. Chem. A, 120, 10072-10087, https://doi.org/10.1021/acs.jpca.6b09370, 2016.
- Molteni, U., Bianchi, F., Klein, F., Haddad, I. E., Frege, C., Rossi, M. J., Dommen, J., and
   Baltensperger, U.: Formation of highly oxygenated organic molecules from aromatic
   compounds, Atmos. Chem. Phys., 18, 1909-1921, https://doi.org/10.5194/acp-18-1909-2018,
   2018.
- Ng, N. L., Kroll, J. H., Chan, A. W. H., Chhabra, P. S., Flagan, R. C., and Seinfeld, J. H.:
   Secondary organic aerosol formation from m-xylene, toluene, and benzene, Atmos. Chem.
   Phys., 7, 3909-3922, https://doi.org/10.5194/acp-7-3909-2007, 2007.
- Nie, W., Yan, C., Huang, D. D., Wang, Z., Liu, Y., Qiao, X., Guo, Y., Tian, L., Zheng, P., Xu, Z., Li,
   Y., Xu, Z., Qi, X., Sun, P., Wang, J., Zheng, F., Li, X., Yin, R., Dallenbach, K. R., Bianchi, F.,
- Pet áj ä, T., Zhang, Y., Wang, M., Schervish, M., Wang, S., Qiao, L., Wang, Q., Zhou, M.,
- Wang, H., Yu, C., Yao, D., Guo, H., Ye, P., Lee, S., Li, Y. J., Liu, Y., Chi, X., Kerminen, V. M.,
   Ehn, M., Donahue, N. M., Wang, T., Huang, C., Kulmala, M., Worsnop, D, Jiang, J., and
- Ding, A.: Secondary organic aerosol formed by condensing anthropogenic vapours over
- China's megacities, Nat. Geosci., 15, 255-261, https://doi.org/10.1038/s41561-022-00922-5, 2022.
- Pankow, J. F., and Asher, W. E.: SIMPOL.1: a simple group contribution method for predicting
   vapor pressures and enthalpies of vaporization of multifunctional organic compounds, Atmos.
   Chem. Phys., 8, 2773-2796, https://doi.org/10.5194/acp-8-2773-2008, 2008.
- Pasik, D., Frandsen, B. N., Meder, M., Iyer, S., Kurt én, T., and Myllys, N.: Gas-phase oxidation of
   atmospherically relevant unsaturated hydrocarbons by acyl peroxy radicals, J. Am. Chem.
   Soc., 146, 13427-13437, https://doi.org/10.1021/jacs.4c02523, 2024.
- Rissanen, M. P., Kurt én, T., Sipil ä, M., Thornton, J. A., Kangasluoma, J., Sarnela, N., Junninen, H.,
  Jørgensen, S., Schallhart, S., Kajos, M. K., Taipale, R., Springer, M., Mentel, T. F.,
- Ruuskanen, T., Pet äj ä, T., Worsnop, D. R., Kjaergaard, H. G., and Ehn, M.: The formation of highly oxidized multifunctional products in the ozonolysis of cyclohexene, J. Am. Chem.
- Soc., 136, 15596-15606, https://doi.org/10.1021/ja507146s, 2014.
- Sebbar, N., Bozzelli, J. W., and Bockhorn, H.: Thermochemistry and reaction paths in the
   oxidation reaction of benzoyl radical: C<sub>6</sub>H<sub>5</sub>C (=O), J. Phys. Chem. A, 115, 11897-11914,
   https://doi.org/10.1021/jp2078067, 2011.
- Shen, H., Vereecken, L., Kang, S., Pullinen, I., Fuchs, H., Zhao, D., and Mentel, T. F.: Unexpected
   significance of a minor reaction pathway in daytime formation of biogenic highly oxygenated
- organic compounds, Sci. Adv., 8, eabp8702, https://doi.org/10.1126/sciadv.abp8702, 2022.

- Sun, J., Wu, F., Hu, B., Tang, G., Zhang, J., and Wang, Y.: VOC characteristics, emissions and
   contributions to SOA formation during hazy episodes, Atmos. Environ., 141, 560-570,
   https://doi.org/10.1016/j.atmosenv.2016.06.060, 2016.
- Tajuelo, M., Bravo, I., Rodr guez, A., Aranda, A., D áz-de-Mera, Y., and Rodr guez, D.:
   Atmospheric sink of styrene, α-methylstyrene, trans-β-methylstyrene and indene: Rate constants and mechanisms of Cl atom-initiated degradation, Atmos. Environ., 200, 78-89, https://doi.org/10.1016/j.atmosenv.2018.11.059, 2019c.
- Tajuelo, M., Rodr guez, A., Baeza-Romero, M. T., Aranda, A., D az-de-Mera, Y., and Rodr guez,
   D.: Secondary organic aerosol formation from α-methylstyrene atmospheric degradation:
   Role of NO<sub>x</sub> level, relative humidity and inorganic seed aerosol, Atmos. Res., 230,
   104631-104640, https://doi.org/10.1016/j.atmosres.2019.104631, 2019b.
- Tajuelo, M., Rodr guez, D., Baeza-Romero, M. T., D áz-de-Mera, Y., Aranda, A., and Rodr guez,
  A.: Secondary organic aerosol formation from styrene photolysis and photooxidation with
  hydroxyl radicals, Chemosphere, 231, 276-286,
  https://doi.org/10.1016/j.chemosphere.2019.05.136, 2019a.
- Wang, H., Ji, Y., Gao, Y., Li, G., and An, T.: Theoretical model on the formation possibility of
   secondary organic aerosol from OH initialed oxidation reaction of styrene in the presence of
   O<sub>2</sub>/NO, Atmos. Environ., 101, 1-9, https://doi.org/10.1016/j.atmosenv.2014.10.042, 2015.
- Wang, L., Wu, R., and Xu, C.: Atmospheric oxidation mechanism of benzene. Fates of alkoxy
  radical intermediates and revised mechanism, J. Phys. Chem. A, 117, 14163-14168,
  https://doi.org/10.1021/jp4101762, 2013.
- Wang, M., Chen, D., Xiao, M., Ye, Q., Stolzenburg, D., Hofbauer, V., Ye, P., Vogel, A. L., Mauldin,
  R. L., Amorim, A., Baccarini, A., Baumgartner, B., Brilke, S., Dada, L., Dias, A., Duplissy, J.,
- Finkenzeller, H., Garmash, O., He, X. C., Hoyle, C. R., Kim, C., Kvashnin, A., Lehtipalo, K.,
- Fischer, L., Molteni, U., Pet äj ä, T., Pospisilova, V., Qu d éver, L. L. J., Rissanen, M., Simon,
- 818 M., Tauber, C., Tom & A., Wagner, A. C., Weitz, L., Volkamer, R., Winkler, P. M., Kirkby, J.,
- Worsnop, D. R., Kulmala, M., Baltensperger, U., Dommen, J., El-Haddad, I., and Donahue,
- N. M.: Photo-oxidation of aromatic hydrocarbons produces low-volatility organic compounds,
   Environ. Sci. Technol., 54, 7911-7921, https://doi.org/10.1021/acs.est.0c02100, 2020.
- Wang, S., and Li, H.: NO<sub>3</sub>-initiated gas-phase formation of nitrated phenolic compounds in polluted atmosphere, Environ. Sci. Technol., 55, 2899-2907, https://doi.org/10.1021/acs.est.0c08041, 2021.
- Wang, S., Newland, M. J., Deng, W., Rickard, A. R., Hamilton, J. F., Muñoz, A., Ródenas, M.,
  Vázquez, M. M., Wang, L., and Wang, X.: Aromatic photo-oxidation, a new source of
  atmospheric acidity, Environ. Sci. Technol., 54, 7798-7806,
  https://doi.org/10.1021/acs.est.0c00526, 2020.
- Wang, S., Wu, R., Berndt, T., Ehn, M., and Wang, L.: Formation of highly oxidized radicals and
   multifunctional products from the atmospheric oxidation of alkylbenzene, Environ. Sci.
   Technol., 51, 8442-8449, https://doi.org/10.1021/acs.est.7b02374, 2017.
- Wu, R., Pan, S., Li, Y., and Wang, L.: Atmospheric oxidation mechanism of toluene, J. Phys.
   Chem. A, 118, 4533-4547, https://doi.org/10.1021/jp500077f, 2014.
- Wu, X., Hou, Q., Huang, J., Chai, J., and Zhang, F.: Exploring the OH-initiated reactions of styrene in the atmosphere and the role of van der Waals complex, Chemosphere, 282, 131004-131012, https://doi.org/10.1016/j.chemosphere.2021.131004, 2021.

https://doi.org/10.5194/egusphere-2025-4646 Preprint. Discussion started: 23 October 2025 © Author(s) 2025. CC BY 4.0 License.

- Wu, X., Huang, C., Niu, S., and Zhang, F.: New theoretical insights into the reaction kinetics of toluene and hydroxyl radicals, Phys. Chem. Chem. Phys., 22, 22279-22288, https://doi.org/10.1039/D0CP02984J, 2020.
- Xu, L., Møller, K. H., Crounse, J. D., Kjaergaard, H. G., and Wennberg, P. O.: New insights into
   the radical chemistry and product distribution in the OH-initiated oxidation of benzene,
   Environ. Sci. Technol., 54, 13467-13477, https://doi.org/10.1021/acs.est.0c04780, 2020.
- Yan, Y., Cabrera-Perez, D., Lin, J., Pozzer, A., Hu, L., Millet, D. B., Porter, W. C., and Lelieveld,
   J.: Global tropospheric effects of aromatic chemistry with the SAPRC-11 mechanism
   implemented in GEOS-Chem version 9-02, Geosci. Model Dev., 12, 111-130,
   https://doi.org/10.5194/gmd-12-111-2019, 2019.
- Yu, S., Jia, L., Xu, Y., and Pan, Y.: Formation of extremely low-volatility organic compounds from
   styrene ozonolysis: Implication for nucleation, Chemosphere, 305, 135459-135467,
   https://doi.org/10.1016/j.chemosphere.2022.135459, 2022.
- Yu, S., Jia, L., Xu, Y., and Pan, Y.: Molecular composition of secondary organic aerosol from
   styrene under different NO<sub>x</sub> and humidity conditions, Atmos. Res., 266, 105950-10604,
   https://doi.org/10.1016/j.atmosres.2021.105950, 2022.
- Zaytsev, A., Koss, A. R., Breitenlechner, M., Krechmer, J. E., Nihill, K. J., Lim, C. Y., Rowe, J. C.,
  Cox, J. L., Moss, J., Roscioli, J. R., Canagaratna, M. R., Worsnop, D. R., Kroll, J. H., and
  Keutsch, F. N.: Mechanistic study of the formation of ring-retaining and ring-opening
  products from the oxidation of aromatic compounds under urban atmospheric conditions,
  Atmos. Chem. Phys., 19, 15117-15129, https://doi.org/10.5194/acp-19-15117-2019, 2019.
- Zhang, H., Wang, J., Dong, B., Xu, F., Liu, H., Zhang, Q., Zong, W., and Shi, X.: New mechanism
  for the participation of aromatic oxidation products in atmospheric nucleation, Sci. Total
  Environ., 917, 170487-170494, https://doi.org/10.1016/j.scitotenv.2024.170487, 2024.
- Zhang, R. M., Truhlar, D. G., and Xu, X.: Kinetics of the toluene reaction with OH radical,
   Research, 2019, Article ID 5373785, https://doi.org/10.34133/2019/5373785, 2019.
- Zhao, H., Zhang, Y., Zhao, Q., Li, Y., and Huang, Z.: A theoretical study of H-abstractions of benzaldehyde by H, O<sup>3</sup>(P), <sup>3</sup>O<sub>2</sub>, OH, HO<sub>2</sub>, and CH<sub>3</sub> radicals: Ab initio rate coefficients and their uncertainty quantification, J. Phys. Chem. A, 126, 7523-7533, https://doi.org/10.1021/acs.jpca.2c02384, 2022.
- Zhao, Y., and Truhlar, D. G.: The M06 suite of density functionals for main group 868 thermochemistry, thermochemical kinetics, noncovalent interactions, excited states, and 869 transition elements: two new functionals and systematic testing of four M06-class functionals 870 12 other functionals, Theor. Chem. Acc., 120, 215-241, 871 https://doi.org/10.1007/s00214-007-0310-x, 2008.