# Peer review of "Mechanistic investigations of the formation of highly oxidized"

_EGUsphere, 2025_

## Author Comment (AC2)

**Comments of reviewer #1**

The manuscript presents a comprehensive and theoretically sound investigation into the multi-generation ·OH oxidation mechanisms of styrene. The study employs high-level quantum chemical calculations to elucidate complex reaction pathways, including the formation of bicyclic peroxy radicals (BPR) and their subsequent reactions, which are crucial for understanding SOA formation from aromatic compounds. The topic is of significant interest to the atmospheric chemistry community. While the computational methodology is robust and the overall conclusions are supported by the data, the manuscript requires revisions to improve clarity, depth of discussion, and to address several specific technical points before it can be accepted for publication.

**Response:** We appreciate the reviewers' comments and have carefully revised the manuscript accordingly. Following is our point-by-point response.

1. please use "OH radical" and "·OH" consistently in the manuscript.

**Response:** Based on the Reviewer's suggestion, "OH radical" and "·OH" have been used throughout the manuscript.

2. In Page 8, when introducing subscript letters (e.g., S2-1-x) to denote species for the first time, please clearly define their meaning.

**Response:** Based on the Reviewer's suggestion, the meaning of subscript letter has been explained in the revised manuscript. The peroxyl radicals $RO_2$ formed from the addition reactions of alkyl radicals R with $O_2$ have multiple possible conformers due to the different orientation of $O_2$ attack. In order to distinguish the different conformers in $RO_2$ radicals, the subscript letter x is used in the present study. The energy ordering of different conformers follows an alphabetical sequence, in which letter a denotes the lowest energy conformer.

Corresponding descriptions have been added in the page 8 line 214-217 of the revised manuscript:

*In order to distinguish the different conformers, the subscript letter x is used in the present study. The energy ordering of different conformers follows an alphabetical sequence, in which letter a denotes the lowest energy conformer.*

3. In Page 8, Line 216, If $\Delta E_r > 59.6$ kcal/mol, this is an endothermic reaction. Please verify whether this description is correct.

**Response:** Based on the Reviewer's suggestion, the value of $\Delta E_r$ has been corrected in the revised manuscript. The attack of an $O_2$ molecule on the C-center site of S1-1 leads to the formation of first generation peroxy radicals S2-1-x ($\Delta E_r > -59.6$ kcal/mol).

Corresponding descriptions have been added in the page 8 line 211-213 of the revised manuscript:

*The attack of an $O_2$ molecule on the C-center site of S1-1 leads to the formation of the first generation peroxy radicals S2-1-x ($\Delta E_r > -59.6$ kcal/mol).*

4. In Page 9, Line 238, when discussing the pseudo-first-order rate constant for the $RO_2$ bimolecular reaction, it is recommended that the authors provide additional clarification regarding the typical atmospheric reactant concentration corresponding to the value of ~0.01 s$^{-1}$. Note that the k value cited in the reference refers to indoor environments.

**Response:** Based on the Reviewer's suggestion, the relevant descriptions on the pseudo-first-order rate constants for the bimolecular reactions of $RO_2$ radicals with $HO_2$ radicals and NO have been added in the revised manuscript. In indoor environments, the concentration of $HO_2$ radicals is ~24 pptv, which is about half of the concentration of NO (Fu et al., 2024). Previous studies have reported that the rate coefficients $k_{RO2+HO2}$ and $k_{RO2+NO}$ for the reactions of alkyl peroxyl radicals with $HO_2$ radicals and NO are $1.7 \times 10^{-11}$ and $9.0 \times 10^{-12}$ cm$^3$ molecule$^{-1}$ s$^{-1}$ (Atkinson and Arey, 2003; Boyd et al., 2003), respectively, leading to the pseudo-first-order rate constants $k'_{RO2+HO2} = k_{RO2+HO2}$ [$HO_2$] and $k'_{RO2+NO} = k_{RO2+NO}$ [NO] of ~0.01 s$^{-1}$ in each case in indoor environments. In the atmosphere, the concentration of $HO_2$ radicals is 20-40 pptv (Bianchi et al., 2019; Wang et al., 2017), resulting in the pseudo-first-order rate constant $k'_{RO2+HO2}$ of 0.01-0.02 s$^{-1}$. The isomerization reaction of $RO_2$ radicals is competitive with the bimolecular reactions with $HO_2$ radicals only when the rate coefficient of intramolecular H-shifts exceeds 0.01-0.02 s$^{-1}$. The typical atmospheric concentration of NO is 0.4-40 ppbv (Bianchi et al., 2019; Wang et al., 2017), leading to the pseudo-first-order rate constant $k'_{RO2+NO}$ of 0.1-10 s$^{-1}$. The intramolecular H-shift reaction of $RO_2$ radicals can compete with the bimolecular reaction with NO only when the rate coefficient of the former case exceeds 10 s$^{-1}$. Therefore, we use the

$k'_{RO2+HO2}$ (0.01-0.02 s$^{-1}$) and $k'_{RO2+NO}$ (0.1-10 s$^{-1}$) values as thresholds to evaluate the relative importance of the isomerization reactions of RO$_2$ radicals under the low- and high-NO$_x$ conditions.

Corresponding descriptions have been added in the page 9 line 251-271 of the revised manuscript:

*In the low-NO$_x$ conditions, the bimolecular reaction with HO$_2$ radicals is expected to be the dominant sink for RO$_2$ radicals (Orlando and Tyndall, 2012; Vereecken et al., 2015). Previous studies have reported that the rate coefficient $k_{RO2+HO2}$ for the reactions of alkyl peroxyl radicals with HO$_2$ radicals is $1.7 \times 10^{-11}$ cm$^3$ molecule$^{-1}$ s$^{-1}$ (Atkinson and Arey, 2003; Boyd et al., 2003). The typical atmospheric concentration of HO$_2$ radicals is 20-40 pptv (Wang et al., 2017; Bianchi et al., 2019), resulting in the pseudo-first-order rate constant $k'_{RO2+HO2} = k_{RO2+HO2}$ [HO$_2$] of 0.01-0.02 s$^{-1}$. The isomerization reaction of RO$_2$ radicals is competitive with the bimolecular reactions with HO$_2$ radicals only when the rate coefficient of intramolecular H-shifts exceeds 0.01-0.02 s$^{-1}$. In the high-NO$_x$ conditions, the bimolecular reaction of RO$_2$ radicals with NO is considered to be a dominant sink (Orlando and Tyndall, 2012; Vereecken et al., 2015). The rate coefficient $k_{RO2+NO}$ for the reaction of alkyl peroxyl radicals with NO is determined to be $9.0 \times 10^{-12}$ cm$^3$ molecule$^{-1}$ s$^{-1}$ (Atkinson and Arey, 2003; Bianchi et al., 2019). The typical atmospheric concentration of NO is 0.4-40 ppbv (Wang et al., 2017; Wang et al., 2019), leading to the pseudo-first-order rate constant $k'_{RO2+NO} = k_{RO2+NO}$ [NO] of 0.1-10 s$^{-1}$. The intramolecular H-shift reaction of RO$_2$ radicals can compete with the bimolecular reaction with NO only when the rate coefficient of the former case exceeds 10 s$^{-1}$. Therefore, we use the $k'_{RO2+HO2}$ (0.01-0.02 s$^{-1}$) and $k'_{RO2+NO}$ (0.1-10 s$^{-1}$) values as thresholds to evaluate the relative importance of the isomerization reactions of RO$_2$ radicals under the low- and high-NO$_x$ conditions.*

5. In Page 11, Line 288, The citation "Zhang et al., 209;" is incomplete and should be corrected to the full and correct reference, which is likely "Zhang et al., 2019".

**Response:** Based on the Reviewer's suggestion, the related reference has been corrected in the revised manuscript.

6. In Page 11, Lines 279-288, authors attribute the discrepancy between this study and the

results of Zhang et al. (2024) at the ·OH addition sites primarily to the latter's failure to account for pre-reaction complexes. This claim requires further evidence to substantiate. It is recommended to supplement the study with complete potential energy surfaces for the C1 and C6 sites under both methodologies. Based on the calculated energy barriers, the branching ratios for ·OH addition to the C1 and C6 sites at 298 K should be computed.

**Response:** Based on the Reviewer's suggestion, the energy barriers ($\Delta E_a$) of all the elementary reactions involved in the addition of OH radical to the different sites of $1^{st}$-ROOH(S4) are recalculated using the DLPNO-CCSD(T)/aug-cc-pVTZ//M06-2X/6-311+G(d,p) method employed in the Zhang's study (Zhang et al., 2024). The calculated energy barriers are compared with those obtained using the M06-2X/6-311++G (3df,3pd)//M06-2X/6-31+g(d,p) method employed in the present study, as shown in Figure 2 and Table S3. In this work, the *syn*-OH-addition is defined as the scenario in which the addition of OH radical occurs at the same side as the −OOH group, while the *anti*-OH-addition is referred to the scenario in which the addition of OH radical occurs at the opposite side as the −OOH group. For the *syn*-OH-addition reactions, the addition of OH radical to the C1-site of $1^{st}$-ROOH (S4) exhibits the lowest barrier ($\Delta E_a$ = 3.6 kcal/mol) due to the stability of the formed product, $P_{S4-add1'}$. A similar conclusion is also obtained from the *anti*-OH-addition reactions that the OH-addition pathway occurring at the C1-site is favorable ($\Delta E_a$ = 0.8 kcal/mol). Notably, the preferred OH-addition pathway in the *anti*-OH-addition reactions exhibits greater competitiveness compared to that in the *syn*-OH-addition reactions. It can be explained by the greater steric hindrance present in the latter reaction.

As shown in Table S3, the $\Delta E_a$ values obtained using the M06-2X/6-311++G(3df,3pd) method are in good agreement with those derived from the DLPNO-CCSD(T)/aug-cc-pVTZ method. The largest deviation and the average absolute deviation are 1.2 and 0.9 kcal/mol, respectively, indicating that the M06-2X/6-311++G(3df,3pd) method employed in this study is reliable. Based on the values of $\Delta E_a$ obtained using the DLPNO-CCSD(T)/ aug-cc-pVTZ method, it can also be concluded that the addition of OH radical to C1-site, occurring at the opposite direction relative to the −OOH group, is energetically favorable. The rate coefficients of the addition of OH radical to the different sites of $1^{st}$-ROOH are calculated to be 8.2 × $10^{-12}$ (C1-site), 5.8 × $10^{-15}$ (C2-site), 8.3 × $10^{-15}$ (C3-site), 8.6 × $10^{-15}$ (C4-site), 2.7 × $10^{-12}$ (C5-site) and 4.1 ×

$10^{-13}$ (C6-site) cm$^3$ molecule$^{-1}$ s$^{-1}$, respectively. The branching ratios for OH addition to the C1, C5 and C6 sites are predicted to be 72.4%, 23.8% and 3.6%, respectively, while the sum of branching ratios for OH addition to other carbon sites is less than 1%.

Our result is opposite to Zhang's finding that the addition of OH radical to C6-site would be the most favorable pathway (Zhang et al., 2024). The discrepancy can be explained by the following three factors: (1) The 1$^{st}$-ROOH conformer selected in the Zhang's study is not the global minimum. In the present study, the global minimum conformer of 1$^{st}$-ROOH, identified through the conformer search, is found to be 2.2 kcal/mol in energy lower than the 1$^{st}$-ROOH structure selected in the Zhang's study (Figure R1). (2) The pre-reactive complexes are not considered in the Zhang's study. The addition of OH radical to C1-, C2-, C3- and C6-sites, occurring at the opposite direction relative to the –OOH group, are merely considered in the Zhang's study. They found that the apparent energy barrier (defined as the energy difference between transition state and reactant) of the addition of OH radical to C6-site is smallest, and is therefore expected to be the favorable pathway. Actually, these OH-addition reactions are modulated by the pre-reactive complexes. It may be inappropriate to determine the favorable pathway based solely on apparent activation energy without considering the pre-reaction complexes. (3) From a geometric perspective, the addition of OH radical to C6-site is associated with greater steric hindrance compared to other sites, as C6-atom connects with a larger functional group. Base on the aforementioned discussions, we believe that the addition of OH radical to C6-site is unlikely to be the dominant pathway. Our calculations also confirm that the addition of OH radical to C6-site is less importance compared to that at the C1-site.

[Figure]

**Figure 2.** PES for the oxidation of 1st-ROOH(S4) initiated by OH radical at the M06-2X/6-311++G(3df,3pd)//M06-2X/6-31+g(d,p) level

[Figure]

**Figure R1.** The geometric parameters and the relative energies ($\Delta E$) of the hydroperoxide 1st-ROOH selected in the Zhang's study and in the present study

**Table S3** The energy barriers ($\Delta E_a$ in kcal/mol) of the addition OH radical to the different sites of 1st-ROOH (S4) calculated at the DLPNO-CCSD(T)/aug-cc-pVTZ//M06-2X/6-311+G(d,p) and M06-2X/6-311++G(3df,3pd)//M06-2X/6-31+g(d,p)levels

| (1st-ROOH (S4)) | DLPNO-CCSD(T)/aug-cc-pVTZ// M06-2X/6-311+G(d,p) | M06-2X/6-311++G(3df,3pd)// M06-2X/6-31+g(d,p) |
|---|---|---|
| *syn*-OH-addition ( OH is added in the same direction of -OOH group) | | |
| C1-site | 3.3 | 3.6 |
| C2-site | 7.8 | 8.8 |

| | | |
|---|---|---|
| C3-site | 8.4 | 9.5 |
| C4-site | 5.4 | 5.7 |
| C5-site | 5.0 | 6.2 |
| C6-site | 4.7 | 5.8 |
| *anti*-OH-addition ( OH is added in the opposite direction of -OOH group) | | |
| C1-site | 2.0 | 0.8 |
| C2-site | 6.3 | 5.3 |
| C3-site | 6.7 | 6.0 |
| C4-site | 5.7 | 4.7 |
| C5-site | 2.6 | 1.4 |
| C6-site | 5.2 | 4.3 |

Corresponding descriptions have been added in the page 12 line 308-356 of the revised manuscript:

*The reaction $1^{st}$-ROOH (S4) + OH proceeds through the addition of OH radical to either side of the benzene ring to yield various alkyl radicals, as depicted in Figure 2. In the present study, syn-OH-addition is defined as the scenario in which the addition of OH radical occurs at the same side as the –OOH group, while anti-OH-addition is referred to the scenario in which the addition of OH radical occurs at the opposite side as the –OOH group. For the syn-OH-addition reactions, the addition of OH radical to the C1-site of $1^{st}$-ROOH (S4) exhibits the lowest barrier ($\Delta E_a$ = 3.6 kcal/mol) due to the stability of the formed product, $P_{S4-add1'}$. A similar conclusion is also obtained from the anti-OH-addition reactions that the OH-addition pathway occurring at the C1-site is favorable ($\Delta E_a$ = 0.8 kcal/mol). Notably, the preferred OH-addition pathway in the anti-OH-addition reactions exhibits greater competitiveness compared to that in the syn-OH-addition reactions. It can be explained by the greater steric hindrance present in the latter reaction. In order to further evaluate the reliability of our results, $\Delta E_a$ of all the syn-OH-addition and anti-OH-addition reactions are recalculated using the DLPNO-CCSD(T)/ aug-cc-pVTZ//M06-2X/6-311+G(d,p) method. As shown in Table S3, the $\Delta E_a$ values obtained using the M06-2X/6-311++G(3df,3pd) method are in good agreement with those derived from the DLPNO-CCSD(T)/aug-cc-pVTZ method. The largest deviation and the average absolute deviation*

*are 1.2 and 0.9 kcal/mol, respectively, indicating that the M06-2X/6-311++G(3df,3pd) method employed in this study is reliable. Based on the values of $\Delta E_a$ obtained using the DLPNO-CCSD(T)/ aug-cc-pVTZ method, it can also be concluded that the addition of OH radical to C1-site, occurring at the opposite direction relative to the –OOH group, is energetically favorable. The rate coefficients of the addition of OH radical to the different sites of $1^{st}$-ROOH are calculated to be $8.2 \times 10^{-12}$ (C1-site), $5.8 \times 10^{-15}$ (C2-site), $8.3 \times 10^{-15}$ (C3-site), $8.6 \times 10^{-15}$ (C4-site), $2.7 \times 10^{-12}$ (C5-site) and $4.1 \times 10^{-13}$ (C6-site) $cm^3$ $molecule^{-1}$ $s^{-1}$, respectively. The branching ratios for OH addition to the C1, C5 and C6 sites are predicted to be 72.4%, 23.8% and 3.6%, respectively, while the sum of branching ratios for OH addition to other carbon sites is less than 1%.*

*Our result is opposite to Zhang's finding that the addition of OH radical to C6-site would be the most favorable pathway (Zhang et al., 2024). The discrepancy can be explained by the following three factors: (1) The $1^{st}$-ROOH conformer selected in the Zhang's study is not the global minimum. In the present study, the global minimum conformer of $1^{st}$-ROOH, identified through the conformer search, is found to be 2.2 kcal/mol lower than the $1^{st}$-ROOH structure selected in the Zhang's study. (2) The pre-reactive complexes are not considered in the Zhang's study. The addition of OH radical to C1-, C2-, C3- and C6-sites, occurring at the opposite direction relative to the –OOH group, are merely considered in the Zhang's study. They found that the apparent energy barrier of the addition of OH radical to C6-site is smallest, and is therefore expected to be the favorable pathway. Actually, these OH-addition reactions are modulated by the pre-reactive complexes. It may be inappropriate to determine the favorable pathway based solely on apparent activation energy without considering the pre-reaction complexes. (3) From a geometric perspective, the addition of OH radical to C6-site is associated with greater steric hindrance compared to other sites, as C6-atom connects with a larger functional group. Base on the aforementioned discussions, we believe that the addition of OH radical to C6-site is unlikely to be the dominant pathway. Our calculations also confirm that the addition of OH radical to C6-site is less importance compared to that at the C1-site.*

7. Why did the author consider only the addition pathway and not the H-abstraction pathway when studying the ·OH oxidation of $2^{nd}$-ROOH (S6) and $2^{nd}$-RONO2 (S26) in Section 3.3?.

**Response:** Based on the Reviewer's suggestion, the H-abstraction pathways involved in the oxidation of $2^{nd}$-ROOH (S6) and $2^{nd}$-RONO$_2$ (S26) initiated by OH radical have been added in the revised manuscript. OH-initiated oxidation of $2^{nd}$-ROOH (S6) can either undergo through the addition of OH radical to either side of the $C_3=C_4$ double bond to generate the alkyl radicals, or proceed via H-abstraction from the different carbon sites to produce the alkyl radicals and alkoxyl radicals, as shown in Figures S16 and S17. For the OH-addition reactions, *syn*-OH-addition is defined as the addition of OH radical on the same side as the –OOH group, while *anti*-OH-addition is referred to the addition of OH radical on the opposite side as the –OOH group. The addition of OH radical to the C3-site of the $C_3=C_4$ double bond forming the product $P_{S6-abs3}$ has the smallest barrier ($\Delta E_a$ = 2.4 kcal/mol) and the exoergicity of -33.5 kcal/mol. For the H-abstraction reactions, the abstraction of hydrogen atom at the $C_5$-site is the most favorable pathway ($\Delta E_a$ = 3.6 kcal/mol) and the exoergicity of -20.2 kcal/mol. It is mainly because that the presence of an allyl group enhances the stability of the resulting product $P_{S6-abs5}$. Notably, the abstraction of hydrogen atom at the $C_2$-site proceeds through a concerted process of $C_2$-H bond and bridge O-O bond rupture, leading to the formation of an alkoxyl radical $P_{S6-abs2}$ ($\Delta E_a$ = 7.2 kcal/mol). This reaction is expected to be less importance due to its higher energy barrier. The rate coefficient of the favorable OH-addition reaction is calculated to be $6.4 \times 10^{-11}$ cm$^3$ molecule$^{-1}$ s$^{-1}$, which is about one order of magnitude greater than that of the preferable H-abstraction reaction ($4.1 \times 10^{-12}$ cm$^3$ molecule$^{-1}$ s$^{-1}$). Based on the above discussion, it can be concluded that OH-addition reaction is favorable on both thermochemically and kinetically. This conclusion is further supported by the OH + alkene reaction systems that OH-addition pathways are predominant (Yang et al., 2017; Chen et al., 2021; Arathala et al., 2024).

OH-initiated oxidation of $2^{nd}$-RONO$_2$ (S26) includes four different OH-addition pathways and five different H-abstraction pathways, as displayed in Figures S19 and S20. For the OH-addition reactions, the attack of OH radical on the C3-site of the $C_3=C_4$ double bond forming the product $P_{S26-add3}$, occurring on the same direction relative to the –ONO$_2$ group, is found to be the favorable pathway ($\Delta E_a$ = 2.4 kcal/mol, $\Delta E_r$ = -33.6 kcal/mol). For the H-abstraction reactions, the abstraction of hydrogen atom at the C5-site is identified as the preferable pathway ($\Delta E_a$ = 5.7 kcal/mol, $\Delta E_r$ = -20.1 kcal/mol) due to the enhanced stability of the resulting product $P_{S26-add5}$ by the presence of an allyl group. By comparing the values of $\Delta E_a$ and $\Delta E_r$ of the favorable

OH-addition and H-abstraction pathways, it can concluded that the former case is dominant on both thermochemically and kinetically. This conclusion is consistent with the result from the reaction $2^{nd}$-ROOH (S6) + OH that OH-addition is more competitive than H-abstraction.

[Figure]

**Figure S16.** PES for the OH-addition reactions involved in the OH-initiated oxidation of $2^{nd}$-ROOH (S6) at the M06-2X/6-311++G(3df,3pd)//M06-2X/6-31+g(d,p) level

[Figure]

**Figure S17.** PES for the H-abstraction reactions involved in the OH-initiated oxidation of $2^{nd}$-ROOH (S6) at the M06-2X/6-311++G(3df,3pd)//M06-2X/6-31+g(d,p) level

[Figure]

**Figure S19.** PES for the OH-addition reactions involved in the OH-initiated oxidation of $2^{nd}$-RONO$_2$ (S26) at the M06-2X/6-311++G(3df,3pd)//M06-2X/6-31+g(d,p) level

[Figure]

**Figure S20.** PES for the H-abstraction reactions involved in the OH-initiated oxidation of $2^{nd}$-RONO$_2$ (S26) at the M06-2X/6-311++G(3df,3pd)//M06-2X/6-31+g(d,p) level

Corresponding descriptions have been added in the page 23 line 589-611 and page 25 line 646-658 of the revised manuscript:

*OH-initiated oxidation of $2^{nd}$-ROOH (S6) can either undergo through the addition of OH radical to either side of the $C_3=C_4$ double bond to generate the alkyl radicals, or proceed via H-abstraction from the different carbon sites to produce the alkyl radicals and alkoxyl radicals, as shown in Figures S16 and S17. For the OH-addition reactions, syn-OH-addition is defined as the*

*addition of OH radical on the same side as the –OOH group, while anti-OH-addition is referred to the addition of OH radical on the opposite side as the –OOH group. The addition of OH radical to the C3-site of the $C_3=C_4$ double bond forming the product $P_{S6-abs3}$ has the smallest barrier ($\Delta E_a$ = 2.4 kcal/mol) and the exoergicity of -33.5 kcal/mol. For the H-abstraction reactions, the abstraction of hydrogen atom at the C5-site is the most favorable pathway ($\Delta E_a$ = 3.6 kcal/mol) and the exoergicity of -20.2 kcal/mol. It is mainly because that the presence of an allyl group enhances the stability of the resulting product $P_{S6-abs5}$. Notably, the abstraction of hydrogen atom at the C2-site proceeds through a concerted process of $C_2$-H bond and -O-O- bridge bond rupture, leading to the formation of an alkoxyl radical $P_{S6-abs2}$ ($\Delta E_a$ = 7.2 kcal/mol). This reaction is expected to be less importance due to its higher energy barrier. The rate coefficient of the favorable OH-addition reaction is calculated to be $6.4 \times 10^{-11}$ $cm^3$ $molecule^{-1}$ $s^{-1}$, which is about one order of magnitude greater than that of the preferable H-abstraction reaction ($4.1 \times 10^{-12}$ $cm^3$ $molecule^{-1}$ $s^{-1}$). Based on the above discussion, it can be concluded that OH-addition reaction is favorable on both thermochemically and kinetically. This conclusion is further supported by the OH + alkene reaction systems that OH-addition pathways are predominant (Chen et al., 2021; Yang et al., 2017; Arathala and Musah, 2024).*

*OH-initiated oxidation of $2^{nd}$-$RONO_2$ (S26) includes four different OH-addition pathways and five different H-abstraction pathways, as displayed in Figures S19 and S20. For the OH-addition reactions, the attack of OH radical on the C3-site of the $C_3=C_4$ double bond forming the product $P_{S26-add3}$, occurring on the same direction relative to the –$ONO_2$ group, is found to be the favorable pathway ($\Delta E_a$ = 2.4 kcal/mol, $\Delta E_r$ = -33.6 kcal/mol). For the H-abstraction reactions, the abstraction of hydrogen atom at the C5-site is identified as the preferable pathway ($\Delta E_a$ = 5.7 kcal/mol, $\Delta E_r$ = -20.1 kcal/mol) due to the enhanced stability of the resulting product $P_{S26-add5}$ by the presence of an allyl group. By comparing the values of $\Delta E_a$ and $\Delta E_r$ of the favorable OH-addition and H-abstraction pathways, it can concluded that the former case is dominant on both thermochemically and kinetically. This conclusion is consistent with the result from the reaction $2^{nd}$-ROOH (S6) + OH that OH-addition is more competitive than H-abstraction.*

Reference

Arathala, P., and Musah, R. A.: Atmospheric chemistry of chloroprene initiated by OH radicals: combined Ab initio/DFT calculations and kinetics analysis, J. Phys. Chem. A, 128, 8983-8995, https://doi.org/10.1021/acs.jpca.4c05428, 2024.

Atkinson, R., and Arey, J.: Atmospheric degradation of volatile organic compounds, Chem. Rev., 103, 4605-4638, https://doi.org/10.1021/cr0206420, 2003.

Bianchi, F., Kurtén, T., Riva, M., Mohr, C., Rissanen, M. P., Roldin, P., Berndt, T., Crounse, J. D., Wennberg, P. O., Mentel, T. F., Wildt, J., Junninen, H., Jokinen, T., Kulmala, M., Worsnop, D. R., Thornton, J. A., Donahue, N., Kjaergaard, H. G., and Ehn, M.: Highly oxygenated organic molecules (HOM) from gas-phase autoxidation involving peroxy radicals: a key contributor to atmospheric aerosol, Chem. Rev., 119, 3472-3509, https://doi.org/10.1021/acs.chemrev.8b00395, 2019.

Boyd, A. A., Flaud, P. M., Daugey, N., and Lesclaux, R.: Rate constants for $RO_2 + HO_2$ reactions measured under a large excess of $HO_2$, J. Phys. Chem. A, 107, 818-821, https://doi.org/10.1021/jp026581r, 2003.

Chen, L., Huang, Y., Xue, Y., Jia, Z., and Wang, W.: Atmospheric oxidation of 1-butene initiated by OH radical: Implications for ozone and nitrous acid formations, Atmos. Environ., 244, 118010-118021, https://doi.org/10.1016/j.atmosenv.2020.118010, 2021.

Fu, Z., Guo, S., Xie, H. B., Zhou, P., Boy, M., Yao, M., and Hu, M.: A near-explicit reaction mechanism of chlorine-initiated limonene: implications for health risks associated with the concurrent use of cleaning agents and disinfectants, Environ. Sci. Technol., 58, 19762-19773, https://doi.org/10.1021/acs.est.4c04388, 2024.

Wang, S., Wu, R., Berndt, T., Ehn, M., and Wang, L.: Formation of highly oxidized radicals and multifunctional products from the atmospheric oxidation of alkylbenzene, Environ. Sci. Technol., 51, 8442-8449, https://doi.org/10.1021/acs.est.7b02374, 2017.

Yang, F., Deng, F., Pan, Y., Zhang, Y., Tang, C., and Huang, Z.: Kinetics of hydrogen abstraction and addition reactions of 3-hexene by OH radicals, J. Phys. Chem. A, 121, 1877-1889, https://doi.org/10.1021/acs.jpca.6b11499, 2017.

Zhang, H., Wang, J., Dong, B., Xu, F., Liu, H., Zhang, Q., Zong, W., and Shi, X.: New mechanism for the participation of aromatic oxidation products in atmospheric nucleation, Sci. Total Environ., 917, 170487-170494, https://doi.org/10.1016/j.scitotenv.2024.170487, 2024.

---

## Author Comment (AC3)

**Comments of reviewer #2**

This study employs theoretical calculations to investigate the reaction mechanism and kinetic properties of the styrene–OH radical reaction. The proposed mechanism is detailed, and the potential link to highly oxygenated molecules (HOMs) is discussed. The work is scientifically sound and contributes valuable insights to the field. However, numerous theoretical studies have previously examined initiation and subsequent reactions (e.g., H-shift, NO reactivity) of benzene-series compounds with OH radicals. The authors should clearly articulate the novelty and advantages of this work compared to existing literature. In addition, the following issues require careful attention and revision.

**Response:** We sincerely thank the reviewers' comments and have carefully revised the manuscript. For the OH-initiated oxidation of benzene-series compounds (e.g., benzene, toluene, benzaldehyde), numerous previous theoretical studies have primarily focused on the first generation OH oxidation mechanisms, including the initiation H-abstraction or OH-addition and subsequent reactions with $O_2$, and the formation mechanisms of the first generation closed-shell products (Zhang et al., 2024; Wu et al., 2020; Xu et al., 2020; Wang et al., 2017 and 2020; Zhang et al., 2019; Iyer et al., 2023; Ji et al., 2017). However, there are few studies on the subsequent oxidation mechanisms of the first generation products, including the multifunctional hydroperoxides, alkoxyl radicals and organic nitrates, which originate from the unimolecular reactions of peroxide bicyclic peroxy radicals (BPR) and their bimolecular reactions with $HO_2$ radicals and NO.

In the present study, the multi-generation OH oxidation mechanisms of styrene are investigated using the quantum chemical methods. For the first generation OH oxidation, the addition of OH radicals to the $C_\beta$-site of vinyl group in styrene is the dominant pathway. The isomerization of the first generation $RO_2$ radicals, formed from the addition reaction of OH-adduct with $O_2$, includes multiple intramolecular H-shift pathways with the rate coefficient $k_{MC\text{-}TST}$ of 1.6 $\times\ 10^{-4}\ s^{-1}$. Among the competing H-shift pathways, the hydrogen atom transfer from the –OH group to the terminal oxygen atom of the –OO group has the lowest barrier. The resulting alkoxy radical decomposes into benzaldehyde through the successive elimination of HCHO and an OH radical. Alternatively, the first generation $RO_2$ radicals can proceed bimolecular reactions with

$HO_2$ radicals and NO to generate the first generation closed-shell C7 and C8-products, benzaldehyde, $1^{st}$-ROOH ($C_8H_{10}O_3$) and $1^{st}$-RONO$_2$ ($C_8H_9NO_3$).

For the second generation OH oxidation, the addition of OH radicals to the *ortho*-position (C1-site) of $1^{st}$-ROOH and $1^{st}$-RONO$_2$, occurring at the opposite direction relative to the –OOH and –ONO$_2$ groups, has the smallest energy barriers. BPR derived from two $O_2$-addition steps and a cyclization process may proceed bimolecular reactions with $HO_2$ radicals and NO to yield the peroxide bicyclic alkoxy radical (BAR), and the second generation closed-shell C8-products, $2^{nd}$-ROOH ($C_8H_{12}O_8$) and $2^{nd}$-RONO$_2$ ($C_8H_{10}N_2O_{10}$) with the fractional yields of 41.4% and 4.8%. The resulting BAR can either proceed the ring-opening reactions to form the dicarbonyl C4-species, or undergo the cyclization reactions to generate the highly oxidized $C_6$-epoxides. For the third generation OH oxidation, the addition of OH radicals to the C3-site of $C_3$=$C_4$ double bond in $2^{nd}$-ROOH and $2^{nd}$-RONO$_2$ is the most favorable pathway. The resulting closed-shell C8-products are the multifunctional hydroperoxides and organic nitrates which possess a peroxide bridge, two carbonyls, two hydroxy and two –OOH or two –NO$_3$ groups. The volatility of the multi-generation OH oxidation products significantly decreases with increasing the number of OH oxidation steps.

1. The manuscript emphasizes HOM formation from the styrene- OH reaction. However, the discussion and conclusions do not clearly identify which products qualify as HOMs. In the mechanism discussion and conclusions, the relationship between the reaction pathway and HOM formation should be more explicitly elaborated, with clear identification of which products qualify as HOMs.

**Response:** Based on the Reviewer's suggestion, the relevant explanations regarding the HOM formation from the multi-generation OH oxidation reactions of styrene have been added in the revised manuscript. The volatility classes for various organic compounds are based on their saturation concentration, as proposed by Donahue et al. (2012). In the present study, the saturated vapour pressure ($P^0$) and saturated concentration ($c^0$) of styrene and its multi-generation OH oxidation closed-shell products are predicted by using the SIMPOL.1 method (Pankow and Asher, 2008). As shown in Table S8, the $P^0$ and $c^0$ of the first generation closed-shell product benzaldehyde ($C_7H_6O$) are $7.62 \times 10^{-4}$ atm and $2.89 \times 10^6$ ug/m$^3$, respectively, which are 3-4

orders of magnitude greater than those of S4 ($C_8H_{10}O_3$, $P^0 = 1.43 \times 10^{-7}$ atm and $c^0 = 8.89 \times 10^2$ ug/m$^3$) and S5 ($C_8H_9NO_3$, $P^0 = 2.54 \times 10^{-7}$ atm and $c^0 = 1.87 \times 10^3$ ug/m$^3$). Based on the values of $c^0$, benzaldehyde is classified as the volatile organic compounds (VOCs), whereas S4 and S5 are classified as the intermediate volatility organic compounds (IVOCs). These first generation closed-shell products exist exclusively in the gas phase under atmospheric conditions (Bianchi et al., 2019).

For the second generation closed-shell products, S6 ($C_8H_{12}O_8$, $c^0 = 4.50 \times 10^{-2}$ ug/m$^3$) and S26 ($C_8H_{10}N_2O_{10}$, $c^0 = 0.18$ ug/m$^3$) formed from the bimolecular reactions with HO$_2$ radicals and NO are classified as the low volatility organic compounds (LVOCs). Similarly, S13 ($C_8H_{10}O_8$, $c^0 = 2.97 \times 10^{-2}$ ug/m$^3$) and S33 ($C_8H_{10}O_8$, $c^0 = 2.97 \times 10^{-2}$ ug/m$^3$), formed through the ring-opening and subsequent intramolecular H-shift reactions of $P_{S4\text{-add3}}$-a-3 and $P_{S5\text{-add3}}$-a-3, respectively, are also classified as LVOCs, which can condense onto the existing large particles (Bianchi et al., 2019). The $c^0$ values of the remaining closed-shell products are significantly greater than those of the aforementioned four products, for example, the $c^0$ values of S20 ($C_6H_8O_6$) and S40-1 ($C_6H_7NO_7$), formed by the cyclization and decomposition reactions of $P_{S4\text{-add3}}$-a-3 and $P_{S5\text{-add3}}$-a-3, are 42.21 and 75.86 ug/m$^3$, respectively, classifying them as the semivolatile organic compounds (SVOC).

For the third generation closed-shell products, the $c^0$ values of S47 ($C_8H_{12}O_9$, $c^0 = 2.68 \times 10^{-4}$ ug/m$^3$) and S51 ($C_8H_{10}O_{10}$, $c^0 = 1.58 \times 10^{-4}$ ug/m$^3$), formed through the O$_2$-addition and subsequent decomposition reactions of $P_{S6\text{-add3}}$, are about two orders of magnitude lower than those of the second generation closed-shell products S6 and S13, despite being classified as LVOCs. Similarly, S58 ($C_8H_{11}NO_{10}$, $c^0 = 5.37 \times 10^{-4}$ ug/m$^3$) and S62 ($C_8H_{10}N_2O_{12}$, $c^0 = 6.18 \times 10^{-4}$ ug/m$^3$), formed via the O$_2$-addition and subsequent decomposition reactions of $P_{S26\text{-add3}}$, exhibit lower $c^0$ values compared to the second generation closed-shell products S26 and S33. The aforementioned results reveal that the volatility of the multi-generation OH oxidation products significantly decreases with increasing the number of OH oxidation steps. As the oxidation reactions of the third generation closed-shell products proceed further, the formed products may possess sufficiently low volatility to participate in the formation and growth of new aerosol particle.

**Table S8** Predicted saturated vapour pressure ($P^0$) and saturated concentrations ($c^0$) of styrene and

| | Species | $P^0$ (atm) | $c^0$ (ug/m$^3$) | Species | $P^0$ (atm) | $c^0$ (ug/m$^3$) |
|---|---|---|---|---|---|---|
| Initial reactant | Styrene ($C_8H_8$) VOC | $4.63 \times 10^{-3}$ | $1.95 \times 10^7$ | | | |
| First generation products | S4 ($C_8H_{10}O_3$) IVOC | $1.43 \times 10^{-7}$ | $8.89 \times 10^2$ | S5 ($C_8H_9NO_3$) IVOC | $2.54 \times 10^{-7}$ | $1.87 \times 10^3$ |
| | Benzaldehyde ($C_7H_6O$) VOC | $7.62 \times 10^{-4}$ | $2.89 \times 10^6$ | | | |
| Second generation products | S6 ($C_8H_{12}O_8$) LVOC | $4.72 \times 10^{-12}$ | $4.50 \times 10^{-2}$ | S10 ($C_4H_4O_2$) VOC | $6.41 \times 10^{-4}$ | $2.17 \times 10^6$ |
| | S10-2 ($C_4H_6O_5$) IVOC | $1.83 \times 10^{-7}$ | $9.91 \times 10^2$ | S13 ($C_8H_{10}O_8$) LVOC | $5.11 \times 10^{-9}$ | $2.97 \times 10^{-2}$ |
| | S14 ($C_4H_4O_2$) VOC | $2.48 \times 10^{-3}$ | $8.39 \times 10^6$ | S20 ($C_6H_8O_6$) SVOC | $5.11 \times 10^{-9}$ | 42.21 |
| | S23 ($C_6H_6O_5$) IVOC | $6.73 \times 10^{-8}$ | $4.31 \times 10^2$ | | | |
| | S26 ($C_8H_{10}N_2O_{10}$) LVOC | $1.52 \times 10^{-11}$ | 0.18 | S30-2 ($C_4H_5NO_6$) IVOC | $3.29 \times 10^{-7}$ | $2.16 \times 10^3$ |

| | | | | | |
|---|---|---|---|---|---|
| | S33 ($C_8H_{10}O_8$) LVOC | $5.11 \times 10^{-9}$ | $2.97 \times 10^{-2}$ | S40-1 ($C_6H_7NO_7$) SVOC | $9.16 \times 10^{-9}$ | 75.86 |

S33 ($C_8H_{10}O_8$)
LVOC
$5.11 \times 10^{-9}$   $2.97 \times 10^{-2}$

S40-1
($C_6H_7NO_7$)
SVOC
$9.16 \times 10^{-9}$   75.86

S40-2 ($C_6H_6O_5$)
IVOC
$6.73 \times 10^{-8}$   $4.31 \times 10^{2}$

Third generation products

S47 ($C_8H_{12}O_9$)
LVOC
$2.64 \times 10^{-14}$   $2.68 \times 10^{-4}$

S48 ($C_6H_8O_7$)
SVOC
$4.63 \times 10^{-10}$   3.59

S51 ($C_8H_{12}O_{10}$)
LVOC
$1.46 \times 10^{-14}$   $1.58 \times 10^{-4}$

($C_8H_{11}NO_{10}$)
LVOC
$4.73 \times 10^{-14}$   $5.37 \times 10^{-4}$

S59 ($C_6H_7NO_8$)
SVOC
$1.14 \times 10^{-9}$   10.16

($C_8H_{10}N_2O_{12}$)
LVOC
$4.70 \times 10^{-14}$   $6.18 \times 10^{-4}$

Corresponding descriptions have been added in the page 28 line 726-762 of the revised manuscript:

*The volatility classes for various organic compounds are based on their saturation concentration, as proposed by Donahue et al. (2012). The saturated vapour pressure ($P^0$) and saturated concentration ($c^0$) of styrene and its multi-generation OH oxidation products are predicted by using the SIMPOL.1 method (Pankow et al., 2008). As show in Table S8, the $P^0$ and $c^0$ of the first generation closed-shell product benzaldehyde ($C_7H_6O$) are $7.62 \times 10^{-4}$ atm and $2.89 \times 10^6$ ug/m$^3$, respectively, which are 3-4 orders of magnitude greater than those of S4 ($C_8H_{10}O_3$, $P^0 = 1.43 \times 10^{-7}$ atm and $c^0 = 8.89 \times 10^2$ ug/m$^3$) and S5 ($C_8H_9NO_3$, $P^0 = 2.54 \times 10^{-7}$ atm and $c^0 = 1.87 \times 10^3$ ug/m$^3$). Based on the values of $c^0$, benzaldehyde is classified as the volatile organic compounds (VOCs), whereas S4 and S5 are classified as the intermediate volatility organic*

*compounds (IVOCs). These first generation closed-shell products exist exclusively in the gas phase under atmospheric conditions (Bianchi et al., 2019).*

*For the second generation closed-shell products, S6 ($C_8H_{12}O_8$, $c^0 = 4.50 \times 10^{-2}$ ug/m$^3$) and S26 ($C_8H_{10}N_2O_{10}$, $c^0 = 0.18$ ug/m$^3$) formed from the bimolecular reactions with $HO_2$ radicals and NO are classified as the low volatility organic compounds (LVOCs). Similarly, S13 ($C_8H_{10}O_8$, $c^0 = 2.97 \times 10^{-2}$ ug/m$^3$) and S33 ($C_8H_{10}O_8$, $c^0 = 2.97 \times 10^{-2}$ ug/m$^3$), formed through the ring-opening and subsequent intramolecular H-shift reactions of $P_{S4-add3}$-a-3 and $P_{S5-add3}$-a-3, respectively, are also classified as LVOCs, which can condense onto the existing large particles (Bianchi et al., 2019). The $c^0$ values of the remaining closed-shell products are significantly greater than those of the aforementioned four products, for example, the $c^0$ values of S20 ($C_6H_8O_6$) and S40-1 ($C_6H_7NO_7$), formed by the cyclization and decomposition reactions of $P_{S4-add3}$-a-3 and $P_{S5-add3}$-a-3, are 42.21 and 75.86 ug/m$^3$, respectively, classifying them as the semivolatile organic compounds (SVOC).*

*For the third generation closed-shell products, the $c^0$ values of S47 ($C_8H_{12}O_9$, $c^0 = 2.68 \times 10^{-4}$ ug/m$^3$) and S51 ($C_8H_{10}O_{10}$, $c^0 = 1.58 \times 10^{-4}$ ug/m$^3$), formed through the $O_2$-addition and subsequent decomposition reactions of $P_{S6-add3}$, are about two orders of magnitude lower than those of the second generation closed-shell products S6 and S13, despite being classified as LVOCs. Similarly, S58 ($C_8H_{11}NO_{10}$, $c^0 = 5.37 \times 10^{-4}$ ug/m$^3$) and S62 ($C_8H_{10}N_2O_{12}$, $c^0 = 6.18 \times 10^{-4}$ ug/m$^3$), formed via the $O_2$-addition and subsequent decomposition reactions of $P_{S26-add3}$, exhibit lower $c^0$ values compared to the second generation closed-shell products S26 and S33. The aforementioned results reveal that the volatility of the multi-generation OH oxidation products significantly decreases with increasing the number of OH oxidation steps. As the oxidation reactions of the third generation closed-shell products proceed further, the formed products may possess sufficiently low volatility to participate in the formation and growth of new aerosol particle.*

2. Explain how low-NO$_x$ and high-NO$_x$ conditions are represented in the computational workflow.

**Response:** Based on the Reviewer's suggestion, the relevant contents have been added in the revised manuscript. In the low-NO$_x$ conditions, the peroxyl radicals RO$_2$ can proceed autoxidation

via sequential intramolecular H-shifts and $O_2$ additions, leading to the formation of highly oxygenated organic molecules (HOMs) (Wang et al., 2017; Rissanen et al., 2014; Iyer et al., 2021). The H-shift reaction serves as the rate-determining step, and therefore limits the formation rate of HOMs in the atmospheric oxidation of volatile organic compounds (VOCs) (Iyer et al., 2021). Moreover, the bimolecular reaction with $HO_2$ radicals is expected to be the dominant sink for $RO_2$ radicals in the low-$NO_x$ conditions (Orlando and Tyndall, 2012; Vereecken et al., 2015). Previous studies have reported that the rate coefficient $k_{RO2+HO2}$ for the reactions of alkyl peroxyl radicals with $HO_2$ radicals is $1.7 \times 10^{-11}$ cm$^3$ molecule$^{-1}$ s$^{-1}$ (Atkinson and Arey, 2003; Boyd et al., 2003). The typical atmospheric concentration of $HO_2$ radicals is 20-40 pptv (Wang et al., 2017; Bianchi et al., 2019), resulting in the pseudo-first-order rate constant $k'_{RO2+HO2} = k_{RO2+HO2}$ [$HO_2$] of 0.01-0.02 s$^{-1}$. The isomerization reaction of $RO_2$ radicals is competitive with the bimolecular reactions with $HO_2$ radicals only when the rate coefficient of intramolecular H-shifts exceeds 0.01-0.02 s$^{-1}$. In the high-$NO_x$ conditions, the bimolecular reaction of $RO_2$ radicals with NO is considered to be a dominant sink (Orlando and Tyndall, 2012; Vereecken et al., 2015). The rate coefficient $k_{RO2+ NO}$ for the reaction of alkyl peroxyl radicals with NO is determined to be $9.0 \times 10^{-12}$ cm$^3$ molecule$^{-1}$ s$^{-1}$ (Bianchi et al., 2019; Atkinson and Arey, 2003). The typical atmospheric concentration of NO is 0.4-40 ppbv (Wang et al., 2017; Bianchi et al., 2019), leading to the pseudo-first-order rate constant $k'_{RO2+NO} = k_{RO2+ NO}$ [NO] of 0.1-10 s$^{-1}$. The intramolecular H-shift reaction of $RO_2$ radicals can compete with the bimolecular reaction with NO only when the rate coefficient of the former case exceeds 10 s$^{-1}$.

To obtain reliable rate coefficients for the intramolecular H-shift reactions of $RO_2$ radicals, the multiconformer transition state theory (MC-TST) is employed in the present study (Møller et al., 2016 and 2020). We use the $k'_{RO2+HO2}$ (0.01-0.02 s$^{-1}$) and $k'_{RO2+NO}$ (0.1-10 s$^{-1}$) values as thresholds to evaluate the relative importance of the isomerization reactions of $RO_2$ radicals under the low- and high-$NO_x$ conditions. Previous studies have also employed the same methodology to evaluate the relative importance of isomerization and bimolecular reactions of $RO_2$ radicals during the OH-initiated oxidation of organophosphate esters and alkylbenzenes (Wang et al., 2017; Fu et al., 2024). For example, for the intramolecular H-shift reactions of the first generation peroxyl radicals S2-1-x, the rate coefficient $k_{MC-TST}$ is calculated to be $1.6 \times 10^{-4}$ s$^{-1}$, which is 2-4 orders of magnitude lower than $k'_{RO2+HO2}$ and $k'_{RO2+NO}$, indicating that the isomerization reaction of S2-1-x is

less competitive than the bimolecular reactions with $HO_2$ radicals and NO.

Corresponding descriptions have been added in the page 9 line 251-278 of the revised manuscript:

*In the low-$NO_x$ conditions, the bimolecular reaction with $HO_2$ radicals is expected to be the dominant sink for $RO_2$ radicals (Orlando and Tyndall, 2012; Vereecken et al., 2015). Previous studies have reported that the rate coefficient $k_{RO2+HO2}$ for the reactions of alkyl peroxyl radicals with $HO_2$ radicals is $1.7 \times 10^{-11}$ $cm^3$ $molecule^{-1}$ $s^{-1}$ (Atkinson and Arey, 2003; Boyd et al., 2003). The typical atmospheric concentration of $HO_2$ radicals is 20-40 pptv (Wang et al., 2017; Bianchi et al., 2019), resulting in the pseudo-first-order rate constant $k'_{RO2+HO2} = k_{RO2+HO2}$ $[HO_2]$ of 0.01-0.02 $s^{-1}$. The isomerization reaction of $RO_2$ radicals is competitive with the bimolecular reactions with $HO_2$ radicals only when the rate coefficient of intramolecular H-shifts exceeds 0.01-0.02 $s^{-1}$. In the high-$NO_x$ conditions, the bimolecular reaction of $RO_2$ radicals with NO is considered to be a dominant sink (Orlando and Tyndall, 2012; Vereecken et al., 2015). The rate coefficient $k_{RO2+NO}$ for the reaction of alkyl peroxyl radicals with NO is determined to be $9.0 \times 10^{-12}$ $cm^3$ $molecule^{-1}$ $s^{-1}$ (Atkinson and Arey, 2003; Bianchi et al., 2019). The typical atmospheric concentration of NO is 0.4-40 ppbv (Wang et al., 2017; Wang et al., 2019), leading to the pseudo-first-order rate constant $k'_{RO2+NO} = k_{RO2+NO}$ $[NO]$ of 0.1-10 $s^{-1}$. The intramolecular H-shift reaction of $RO_2$ radicals can compete with the bimolecular reaction with NO only when the rate coefficient of the former case exceeds 10 $s^{-1}$. Therefore, we use the $k'_{RO2+HO2}$ (0.01-0.02 $s^{-1}$) and $k'_{RO2+NO}$ (0.1-10 $s^{-1}$) values as thresholds to evaluate the relative importance of the isomerization reactions of $RO_2$ radicals under the low- and high-$NO_x$ conditions. Previous studies have also employed the same methodology to evaluate the relative importance of isomerization and bimolecular reactions of $RO_2$ radicals during the OH-initiated oxidation of organophosphate esters and alkylbenzenes (Wang et al., 2017; Fu et al., 2024). For the intramolecular H-shift reactions of S2-1-x, the rate coefficient $k_{MC-TST}$ is estimated to be $1.6 \times 10^{-4}$ $s^{-1}$, which is 2-4 orders of magnitude lower than $k'_{RO2+HO2}$ and $k'_{RO2+NO}$, indicating that the isomerization reaction of S2-1-x is less competitive than the bimolecular reactions with $HO_2$ radicals and NO.*

3. Current kinetic interpretation relies heavily on potential energy barriers without considering other factors such as pre-exponential terms and tunneling effects. To substantiate

claims, calculate rate constants for key competing steps. Statements like "Based on ΔEa, H-shift is the rate-determining step" are speculative without quantitative kinetic data. Re-examine all sections where competitiveness is inferred without rate constant calculations.

**Response:** Based on the Reviewer's suggestion, the rate coefficients for all key competing reactions have been calculated in the revised manuscript. In the present study, the rate coefficients of unimolecular reactions, including intramolecular H-shifts, cyclization, $HO_2$-elimination, and C-C bond and C-O bond scissions, are calculated using the RRKM theory along with energy-grained master equation (RRKM-ME) (Holbrook et al., 1996). The rate coefficients of bimolecular reactions, involving H-abstraction and OH-addition, are determined using the traditional transition state theory (TST) (Fernández-Ramos et al., 2007). An asymmetric one-dimensional Eckart model (Eckart, 1930) is employed to consider the tunneling correction factors in the rate coefficient calculations based on RRKM-ME and TST. For the intramolecular H-shift reactions of $RO_2$ and RO radicals, the rate coefficients are calculated employing the multiconformer transition state theory (MC-TST) (Møller et al., 2016 and 2020).

As depicted in Figure 1, the intramolecular H-shift reactions of the first generation peroxyl radical S2-1-x proceed through the hydrogen atom transfer from the $-CH_2$, $-CH$ and $-OH$ groups to the terminal oxygen atom of the $-OO$ group to yield various alkyl and alkoxyl radicals. Among these competing H-shift reactions, the hydrogen atom at the $-OH$ group can be transferred via a six-membered ring transition state (1,5-H shift) to yield an alkoxyl radical S3-1-a, which exhibits the lowest barrier ($\Delta E_a = 21.0$ kcal/mol). The resulting S3-1-a can undergo the $C_\alpha$-$C_\beta$ bond cleavage to produce a formaldehyde and an alkyl radical S4-1-a ($\Delta E_a = 0.8$ kcal/mol), followed by an OH radical release to form benzaldehyde ($\Delta E_a = 0.1$ kcal/mol). The rate coefficients for the aforementioned three pathways are calculated to be $2.7 \times 10^{-4}$, $4.6 \times 10^{10}$ and $7.2 \times 10^{10}$ s$^{-1}$, respectively. Based on the values of $\Delta E_a$ and the corresponding rate coefficients, it can be concluded that the 1,5-H shift reaction is the rate-determining step in the formation of benzaldehyde.

As depicted in Figure S18, the unimolecular decay of the product $P_{S6-add3}$ resulting from the favorable OH-addition pathway in the reaction OH + 2$^{nd}$-ROOH (S6) proceeds through a cyclization process to yield an epoxide compound S41 and an OH radical byproduct with the $\Delta E_a$ of 15.3 kcal/mol and the rate coefficient $k_{R41}$ of $1.8 \times 10^2$ s$^{-1}$, or undergoes via intramolecular 1,4

H-shift to form a peroxy radical S42 with the $\Delta E_a$ of 21.8 kcal/mol and the rate coefficient $k_{R42}$ of 1.9 $s^{-1}$, or proceeds via the elimination of hydrogen atom to produce an alkene S43 with the $\Delta E_a$ of 37.9 kcal/mol. Based on the values of $\Delta E_a$ and the corresponding rate coefficients, the dominant pathway of the unimolecular decomposition of $P_{S6-add3}$ is the formation of S41.

The product $P_{S26-add3}$ arising from the favorable OH-addition pathway in the reaction OH + $2^{nd}$-RONO$_2$ (S26) has three potential unimolecular decay pathways, as depicted in Figure S21: (1) $P_{S26-add3}$ dissociates to an epoxide S52 and a NO$_2$ molecule through a cyclization process with the $\Delta E_a$ of 18.5 kcal/mol and the rate coefficient $k_{R52}$ of 0.4 $s^{-1}$; (2) $P_{S26-add3}$ isomerizes to an alkyl radical S53 via the intramolecular 1,2 H-shift ($\Delta E_a$ = 40.0 kcal/mol); (3) $P_{S26-add3}$ converts into an alkene S54 via the elimination of hydrogen atom ($\Delta E_a$ = 39.1 kcal/mol). Based on the value of $\Delta E_a$ and the corresponding rate coefficient, the dominant pathway of the unimolecular decomposition of $P_{S26-add3}$ is the formation of S52.

[Figure]

**Figure 1.** PES for the first-stage oxidation of styrene initiated by OH radicals and the isomerization reactions of S2-1-x at the M06-2X/6-311++G(3df,3pd)//M06-2X/6-31+g(d,p) level

[Figure]

**Figure S18.** PES for the unimolecular decomposition of $P_{S6\text{-}add3}$ at the M06-2X/6-311++G(3df,3pd)//M06-2X/6-31+g(d,p) level

[Figure]

**Figure S21.** PES for the unimolecular decomposition of $P_{S26\text{-}add3}$ at the M06-2X/6-311++G(3df,3pd)//M06-2X/6-31+g(d,p) level

Corresponding descriptions have been added in the page 8 line 219-231, page 24 line 612-620, and page 26 line 659-667 of the revised manuscript:

*For the unimolecular decomposition reactions of S2-1-x, there are three kinds of pathways. One is the intramolecular H-shift reactions, where the hydrogen atom migrates from the $-CH_2$, $-CH$ and $-OH$ groups to the terminal oxygen atom of the $-OO$ group leading to various alkyl and alkoxyl radicals. Among these competing H-shift reactions, the hydrogen atom at the $-OH$ group can be transferred via a six-membered ring transition state (1,5-H shift) to yield an alkoxyl radical*

*S3-1-a, which exhibits the lowest barrier ($\Delta E_a$ = 21.0 kcal/mol). The resulting S3-1-a can undergo the $C_\alpha$-$C_\beta$ bond cleavage to produce a formaldehyde and an alkyl radical S4-1-a ($\Delta E_a$ = 0.8 kcal/mol), followed by an OH radical release to form benzaldehyde ($\Delta E_a$ = 0.1 kcal/mol). The rate coefficients for the aforementioned three pathways are calculated to be $2.7 \times 10^{-4}$, $4.6 \times 10^{10}$ and $7.2 \times 10^{10}$ $s^{-1}$, respectively. Based on the values of $\Delta E_a$ and the corresponding rate coefficients, it can be concluded that the 1,5-H shift reaction is the rate-determining step in the formation of benzaldehyde.*

*As depicted in Figure S18, the unimolecular decay of the product $P_{S6-add3}$ resulting from the favorable OH-addition reaction proceeds through a cyclization process to yield an epoxide compound S41 and an OH radical byproduct with the $\Delta E_a$ of 15.3 kcal/mol and the rate coefficient $k_{R41}$ of $1.8 \times 10^2$ $s^{-1}$, or undergoes via intramolecular 1,4 H-shift to form a peroxy radical S42 with the $\Delta E_a$ of 21.8 kcal/mol and the rate coefficient $k_{R43}$ of 1.9 $s^{-1}$, or proceeds via the elimination of hydrogen atom to produce an alkene S43 with the $\Delta E_a$ of 37.9 kcal/mol. Based on the values of $\Delta E_a$ and the corresponding rate coefficients, the dominant pathway of the unimolecular decomposition of $P_{S6-add3}$ is the formation of S41.*

*The product $P_{S26-add3}$ arising from the favorable OH-addition pathway has three potential unimolecular decay pathways, as depicted in Figure S21: (1) $P_{S26-add3}$ dissociates to an epoxide S52 and a $NO_2$ molecule through a cyclization process with the $\Delta E_a$ of 18.5 kcal/mol and the rate coefficient $k_{R52}$ of 0.4 $s^{-1}$; (2) $P_{S26-add3}$ isomerizes to an alkyl radical S53 via the intramolecular 1,2 H-shift ($\Delta E_a$ = 40.0 kcal/mol); (3) $P_{S26-add3}$ converts into an alkene S54 via the elimination of hydrogen atom ($\Delta E_a$ = 39.1 kcal/mol). Based on the value of $\Delta E_a$ and the corresponding rate coefficient, the dominant pathway of the unimolecular decomposition of $P_{S26-add3}$ is the formation of S52.*

4. While individual product generations are discussed, the manuscript lacks an integrated view of the overall mechanism and product yields. It is strongly recommended that the authors leverage their MESMER 6.0 kinetic model to simulate and present the branching ratios (or fractional yields) of the major products. Present fractional yields for major products and illustrate competition between $RO_2$ + NO and $RO_2$ + $HO_2$ channels, highlighting how pathway dominance shifts with $NO_x$ levels.

**Response:** Based on the Reviewer's suggestion, the overall reaction mechanism and the fractional yields of the major products in the multi-generation OH oxidation of styrene have been added in the revised manuscript. As shown in Figure S22, in the low-$NO_x$ conditions, the fractional yield of the first generation closed-shell product $1^{st}$-ROOH (S4) formed from the reaction S2-1-x + $HO_2$ · is predicted to be 71.6%. For the second generation OH oxidation, the reaction of the peroxyl radical $P_{S4-add3}$-a-2 with $HO_2$ radicals produces the second generation closed-shell product $2^{nd}$-ROOH (S6) and an alkoxyl radical $P_{S4-add3}$-a-3, with the fractional yields of 41.4% and 10.4%, respectively. The formed $P_{S4-add3}$-a-2 can either proceed through the $C_5$-$C_6$ bond scission to produce an alkyl radical S7 with the fractional yield of 7.8%, or undergo via a cyclization process to generate an alkyl radical S15 with the fractional yield of 2.6%. S7 and S15 can be transformed via a series of reactions, ultimately leading to the formation of second generation closed-shell product S10-2, S13 and S23, with the fractional yields of 5.6%, 2.2% and 1.3%, respectively. For the third generation OH oxidation, the degradation of $2^{nd}$-ROOH (S6) ultimately yields the third generation closed-shell products S47 and S51, with the fractional yields of 26.3% and 0.3%, respectively. As a result, the major closed-shell products are $1^{st}$-ROOH (S4), $2^{nd}$-ROOH (S6), S10-2, S13, and S47 in the multi-generation OH oxidation of styrene in the low-$NO_x$ conditions.

In the high-$NO_x$ conditions, the fractional yield of the first generation closed-shell product $1^{st}$-$RONO_2$ (S5) formed from the reaction S2-1-x + NO is predicted to be 26.5%, as shown in Figure S23. As the OH oxidation reactions proceed, $1^{st}$-$RONO_2$ (S5) can be initially transformed into the peroxyl radical $P_{S5-add3}$-a-2, followed by reaction with NO to form the second generation closed-shell product $2^{nd}$-$RONO_2$ (S26) and an alkoxyl radical $P_{S5-add3}$-a-3, with the fractional yields of 4.8% and 11.2%, respectively. The decomposition of $P_{S5-add3}$-a-3 undergoes via two distinct pathways. One is the $C_5$-$C_6$ bond cleavage, leading to an alkyl radical S27 with the fractional yield of 7.8%. The other is the cyclization, resulting in an alkyl radical S35 with the fractional yield of 3.4%. The resulting S27 and S35 undergo multiple oxidation steps, finally leading to the formation of the second generation closed-shell products S30-2, S33 and S40-1, with the fractional yields of 6.0%, 1.8%, and 1.7%, respectively. $2^{nd}$-$RONO_2$ (S26) can be further oxidized to yield the third generation closed-shell products S58 and S62, with the fractional yields of 2.6% and 0.03%, respectively. In summary, the major closed-shell products are $1^{st}$-$RONO_2$ (S5),

$2^{nd}$-RONO$_2$ (S26), S30-2 and S58 in the multi-generation OH oxidation of styrene in the high-NO$_x$ conditions.

**Figure S22.** Overall reaction mechanism of the multi-generation OH oxidation of styrene in the low-NO$_x$ conditions. The blue, green, and pink molecular structures represent the first, second and third generation closed-shell products, respectively. The rate coefficients are displayed in red. The calculated fractional yields of the closed-shell products are represented in blue. The reaction of peroxyl radical with HO$_2$ radicals is assumed to yield ROOH with a 80% branching ratio and RO

radicals with a 20% branching ratio, based on the reaction of OH-substituted peroxy species with HO$_2$ radicals (Orlando and Tyndall, 2012).

**Figure S23.** Overall reaction mechanism of the multi-generation OH oxidation of styrene in the high-NO$_x$ conditions. The blue, green, and pink molecular structures represent the first, second and third generation closed-shell products, respectively. The rate coefficients are displayed in red. The calculated fractional yields of the closed-shell products are represented in blue. The reaction of peroxyl radical with NO is assumed to yield RONO$_2$ with a 30% branching ratio and RO radicals with a 70% branching ratio, based on the reaction of peroxy species containing 10 carbon atoms with NO (Orlando and Tyndall, 2012).

Corresponding descriptions have been added in the page 27 line 687-721 of the revised manuscript:

*The overall reaction mechanism and the fractional yields of the major products in the multi-generation OH oxidation of styrene under different $NO_x$ conditions are presented in Figures S22 and S23. In the low-$NO_x$ conditions, the fractional yield of the first generation closed-shell product $1^{st}$-ROOH (S4) formed from the reaction S2-1-x + $HO_2$ ·is predicted to be 71.6%. For the second generation OH oxidation, the reaction of the peroxyl radical $P_{S4\text{-}add3}$-a-2 with $HO_2$ radicals produces the second generation closed-shell product $2^{nd}$-ROOH (S6) and an alkoxyl radical $P_{S4\text{-}add3}$-a-3, with the fractional yields of 41.4% and 10.4%, respectively. The formed $P_{S4\text{-}add3}$-a-2 can either proceed through the $C_5$-$C_6$ bond scission to produce an alkyl radical S7 with the fractional yield of 7.8%, or undergo via a cyclization process to generate an alkyl radical S15 with the fractional yield of 2.6%. S7 and S15 can be transformed via a series of reactions, ultimately leading to the formation of second generation closed-shell product S10-2, S13 and S23, with the fractional yields of 5.6%, 2.2% and 1.3%, respectively. For the third generation OH oxidation, the degradation of $2^{nd}$-ROOH (S6) ultimately yields the third generation closed-shell products S47 and S51, with the fractional yields of 26.3% and 0.3%, respectively. As a result, the major closed-shell products are $1^{st}$-ROOH (S4), $2^{nd}$-ROOH (S6), S10-2, S13 and S47 in the multi-generation OH oxidation of styrene in the low-$NO_x$ conditions.*

*In the high-$NO_x$ conditions, the fractional yield of the first generation closed-shell product $1^{st}$-$RONO_2$ (S5) formed from the reaction S2-1-x + NO is predicted to be 26.5%, as shown in Figure S23. As the OH oxidation reactions proceed, $1^{st}$-$RONO_2$ (S5) can be initially transformed into the peroxyl radical $P_{S5\text{-}add3}$-a-2, followed by reaction with NO to form the second generation closed-shell product $2^{nd}$-$RONO_2$ (S26) and an alkoxyl radical $P_{S5\text{-}add3}$-a-3, with the fractional yields of 4.8% and 11.2%, respectively. The decomposition of $P_{S5\text{-}add3}$-a-3 undergoes via two distinct pathways. One is the $C_5$-$C_6$ bond cleavage, leading to an alkyl radical S27 with the fractional yield of 7.8%. The other is the cyclization, resulting in an alkyl radical S35 with the fractional yield of 3.4%. The resulting S27 and S35 undergo multiple oxidation steps, finally leading to the formation of the second generation closed-shell products S30-2, S33 and S40-1, with the fractional yields of 6.0%, 1.8%, and 1.7%, respectively. $2^{nd}$-$RONO_2$ (S26) can be further oxidized to yield the third generation closed-shell products S58 and S62, with the fractional yields*

*of 2.6% and 0.03%, respectively. In summary, the major closed-shell products are 1ˢᵗ-RONO₂ (S5),*

*2ⁿᵈ-RONO₂ (S26), S30-2 and S58 in the multi-generation OH oxidation of styrene in the high-NOₓ*

*conditions.*

5. The discussion regarding the reaction pathways for certain intermediates, such as S9 in Figure 3 and S29 in Figure 5… appears to be incomplete. Specifically, these radical intermediates are theoretically capable of undergoing additional reactions, such as H-atom shifts or $O_2$ addition, which are not depicted. The authors should either clarify the rationale for excluding these pathways (e.g., based on kinetic insignificance) or include them in the mechanism for a more comprehensive analysis.

**Response:** Based on the Reviewer's suggestion, the addition reactions S9 + $O_2$ and S29 + $O_2$, along with the subsequent intramolecular H-shift reactions have been added in the revised manuscript. The unimolecular decomposition of S9 can proceed through the C1-C2 bond scission to produce a ketene-enol S10 and an alkyl radical S10-1 ($\Delta E_a$ = 16.1 kcal/mol), followed by reaction with $O_2$ leading to a $HO_2$ radical and a 1,2-dicarbonyl compound S10-2 ($\Delta E_a$ = 14.0 kcal/mol). Alternatively, S9 may undergo via the elimination of CO to generate an alkyl radical S11 ($\Delta E_a$ = 29.4 kcal/mol). The aforementioned results show that the formation of S10 and S10-1 is energetically favorable, with the rate coefficient $k_{S10}$ of 26.1 s$^{-1}$. In the presence of $O_2$, the attack of an $O_2$ molecule on the C-centered site of S9 leads to the fourth generation peroxyl radical S12-x ($\Delta E_r$ > -20.5 kcal/mol). Adopting the rate coefficient $k_{R+O2}$ of 6.0 ×10$^{-12}$ cm$^3$ molecule$^{-1}$ s$^{-1}$ for the reactions of alkyl radicals with $O_2$, and the atmospheric $O_2$ concentration of 5 × 10$^{18}$ molecule cm$^{-3}$ (Ma et al., 2021), the pseudo-first-order rate constant $k'_{R+O2}$ = $k_{R+O2}$ [$O_2$] is 3.0 × 10$^7$ s$^{-1}$. $k'_{R+O2}$ is about six orders of magnitude greater than $k_{S10}$, indicating that the unimolecular decomposition of S9 is less importance. As shown in Figure S9, S12-x can proceed various intramolecular H-shift reactions, where hydrogen atom migrates from the different carbon sites or hydroxyl groups to the terminal oxygen atom of the –OO group, resulting in the formation of QOOH radicals and alkoxyl radicals. Among these competing H-shift reactions, the 1,7-H transfer at the Cα-site leading to the formation of S12-d-P exhibits the smallest barrier ($\Delta E_a$ = 17.4 kcal/mol). Then, it decomposes to yield an OH radical and a closed-shell product S13 containing a hydroperoxide, three hydroxyl and three carbonyl groups ($\Delta E_a$ =1.1 kcal/mol).

The unimolecular decomposition of S29 proceeds via two distinct pathways. One is the C1-C2 bond cleavage, yielding a ketene-enol S30 and an alkyl radical S30-1 ($\Delta E_a$ = 17.8 kcal/mol), followed by reaction with $O_2$ to form a $HO_2$ radical and a 1,2-dicarbonyl species S30-2 ($\Delta E_a$ = 11.7 kcal/mol). The other is the elimination of CO to generate an alkyl radical S31 ($\Delta E_a$ = 24.8 kcal/mol), but the barrier is considerably high, making this pathway less competitive. The rate coefficient for the formation of S30 and S30-1 is calculated to be 14.4 $s^{-1}$, which is about six orders of magnitude lower than the pseudo-first-order rate constant $k'_{R+O2}$, indicating that the unimolecular decomposition of S29 is insignificant. In the presence of $O_2$, the bimolecular reaction of S29 with $O_2$ produces the fourth generation peroxyl radicals S32-x, comprising five energetically similar conformers as shown in Figure S14. For the 1,7-H transfer reaction, hydrogen atom at the Cα-site can be transferred through an eight-membered ring transition state to generate an alkyl radical S32-d-P ($\Delta E_a$ = 23.3 kcal/mol), followed by the elimination of $NO_2$ to form a closed product S33 ($\Delta E_a$ = 1.0 kcal/mol). S33 and S13 are isomeric species, with the former exhibiting more stability than the latter.

**Figure S9.** PES for the intramolecular hydrogen transfer reactions of S12-x at the M06-2X/6-311++G(3df,3pd)//M06-2X/6-31+g(d,p) level

**Figure S14.** PES for the intramolecular hydrogen transfer reactions of S32-x at the M06-2X/6-311++G(3df,3pd)//M06-2X/6-31+g(d,p) level

Corresponding descriptions have been added in the page 16 line 443-466 and page 20 line 535-551 of the revised manuscript:

*The unimolecular decomposition of S9 can proceed through the C1-C2 bond scission to produce a ketene-enol S10 and an alkyl radical S10-1 ($\Delta E_a = 16.1$ kcal/mol), followed by reaction with $O_2$ leading to a $HO_2$ radical and a 1,2-dicarbonyl compound S10-2 ($\Delta E_a = 14.0$ kcal/mol). Alternatively, S9 may undergo via the elimination of CO to generate an alkyl radicals S11 ($\Delta E_a = 29.4$ kcal/mol). The aforementioned results show that the formation of S10 and S10-1 is energetically favorable, with the rate coefficient $k_{S10}$ of 26.1 $s^{-1}$.*

*In the presence of $O_2$, the attack of an $O_2$ molecule on the C-centered sites of S9 leads to the fourth generation peroxyl radical S12-x ($\Delta E_r > -20.5$ kcal/mol). Adopting the rate coefficient $k_{R+O2}$ of $6.0 \times 10^{-12}$ $cm^3$ $molecule^{-1}$ $s^{-1}$ for the reactions of alkyl radicals with $O_2$, and the atmospheric $O_2$ concentration of $5 \times 10^{18}$ molecule $cm^{-3}$ (Ma et al., 2021), the pseudo-first-order rate constant $k'_{R+O2} = k_{R+O2}$ $[O_2]$ is $3.0 \times 10^7$ $s^{-1}$. The unimolecular decomposition of alkyl radicals is competitive only when their decay rate exceeds $3.0 \times 10^7$ $s^{-1}$. $k'_{R+O2}$ is about six orders of magnitude greater than $k_{S10}$, indicating that the unimolecular decomposition of S9 is less importance. As shown in Figure S9, S12-x can proceed various intramolecular H-shift reactions,*

*where hydrogen atom migrates from the different carbon sites or hydroxyl groups to the terminal oxygen atom of the –OO group, resulting in the formation of QOOH radicals and alkoxyl radicals. Among these competing H-shift reactions, the 1,7-H transfer at the Cα-site leading to the formation of S12-d-P exhibits the smallest barrier ($\Delta E_a$ = 17.4 kcal/mol). Then, it decomposes to yield an OH radical and a closed-shell product S13 containing a hydroperoxide, three hydroxyl and three carbonyl groups ($\Delta E_a$ = 1.1 kcal/mol).*

*S28-e-P can readily isomerize to S29, which includes two distinct decomposition pathways. One is the C1-C2 bond cleavage, yielding a ketene-enol S30 and an alkyl radical S30-1 ($\Delta E_a$ = 17.8 kcal/mol), followed by reaction with $O_2$ to form a $HO_2$ radical and a 1,2-dicarbonyl species S30-2 ($\Delta E_a$ = 11.7 kcal/mol). The other is the elimination of CO to generate an alkyl radical S31 ($\Delta E_a$ = 24.8 kcal/mol), but the barrier is considerably high, making this pathway less competitive. The rate coefficient for the formation of S30 and S30-1 is calculated to be 14.4 $s^{-1}$, which is about six orders of magnitude lower than the pseudo-first-order rate constant $k'_{R+O2}$, indicating that the unimolecular decomposition of S29 is insignificant.*

*In the presence of $O_2$, the bimolecular reaction of S29 with $O_2$ produces the fourth generation peroxyl radicals S32-x, comprising five energetically similar conformers as shown in Figure S14. For the 1,7-H transfer reaction, hydrogen atom at the Cα-site can be transferred through an eight-membered ring transition state to generate an alkyl radical S32-d-P ($\Delta E_a$ = 23.3 kcal/mol), followed by the elimination of $NO_2$ forming a closed product S33 ($\Delta E_a$ = 1.0 kcal/mol). S33 and S13 are isomeric species, with the former exhibiting more stability than the latter.*

6. Methodology (Lines 135-139): Given the relatively small reaction system, CCSD(T) calculations are computationally feasible. Benchmarking the adopted methods for this specific system is essential to validate accuracy.

**Response:** Based on the Reviewer's suggestion, the single point energies of all the stationary points involved in the initial addition of OH radicals to styrene and intramolecular H-shift reactions of first generation peroxyl radicals S2-1-x are recalculated using the DLPNO-CCSD(T)/ aug-cc-pVTZ method performed using the Orca 6.1 program. The calculated energy barriers ($\Delta E_a$) are compared with the values derived from the M06-2X/6-311++G(3df,3pd) method employed in the present study. As shown in Table S1, the $\Delta E_a$ values obtained using the

M06-2X/6-311++G(3df,3pd) method are consistent with those derived from the DLPNO-CCSD(T)/aug-cc-pVTZ method. The largest deviation and the average absolute deviation are 1.2 and 0.6 kcal/mol, respectively, indicating that the computational method employed in this study is reliable. Considering the computational cost, the M06-2X/6-311++G(3df,3pd) method is employed to investigate the formation mechanism of highly oxidized products from the multi-generation OH oxidation of styrene.

**Table S1** The energy barriers ($\Delta E_a$ in kcal/mol) of all the elementary reactions involved in the initial addition of OH radicals to styrene and intramolecular H-shift reactions of first generation peroxyl radicals S2-1-x calculated at the DLPNO-CCSD(T)/aug-cc-pVTZ and M06-2X/6-311++G(3df,3pd) levels

| Entry | DLPNO-CCSD(T)/aug-cc-pVTZ | M06-2X/6-311++G(3df,3pd) |
|---|---|---|
| $R_{TS1}$ | 0.4 | 0.8 |
| $R_{TS2-1-a}$ | 21.5 | 21.0 |
| $R_{TS2-1-b}$ | 32.4 | 33.3 |
| $R_{TS2-1-c}$ | 32.6 | 33.6 |
| $R_{TS2-1-d}$ | 27.5 | 27.9 |
| $R_{TS2-1-e}$ | 30.1 | 31.3 |
| $R_{TS2-1-f}$ | 31.0 | 31.1 |
| $R_{TS2-1-h}$ | 30.5 | 30.6 |
| $R_{TS3-1-a}$ | 1.3 | 0.8 |

Corresponding descriptions have been added in the page 5 line 137-148 of the revised manuscript:

*In order to further evaluate the reliability of the computational method employed herein, the single point energies of all the stationary points involved in the initial addition of OH radicals to styrene and intramolecular H-shift reactions of the first generation peroxyl radicals S2-1-x are recalculated using the DLPNO-CCSD(T)/ aug-cc-pVTZ method performed using the Orca 6.1 program (Neese, 2025). As shown in Table S1, the $\Delta E_a$ values obtained using the M06-2X/6-311++G(3df,3pd) method are consistent with those derived from the DLPNO-CCSD(T)/aug-cc-pVTZ method. The largest deviation and the average absolute deviation are 1.2 and 0.6 kcal/mol, respectively, indicating that the computational method employed in this study is reliable. Considering the computational cost, the M06-2X/6-311++G(3df,3pd) method is*

*employed to investigate the formation mechanism of highly oxidized products from the multi-generation OH oxidation of styrene.*

7. Section 2.2, Please clarify why conformation searches were specifically performed for $RO_2 \cdot$ and $RO \cdot$ species. How were the initial conformer samples selected?

**Response:** Based on the Reviewer's suggestion, the relevant explanations on the conformer search of $RO_2$ and $RO$ radicals have been added in the revised manuscript. $RO_2$ radicals formed from the addition of $O_2$ to the carbon-centered site of alkyl radicals R have multiple possible conformers due to the different orientations of $O_2$ attack. An initial structure of $RO_2$ radicals is optimized at the B3LYP/6-31+G(d) level and subsequently used as the starting geometry to perform the conformer search conducted using the Molclus program. The resulting structures are initially optimized at the B3LYP/6-31+G(d) level, as this method accurately predicts the relative energy ordering of different conformers (Møller et al., 2016 and 2020). All unique conformers within 5.0 kcal/mol with respect to the lowest energy conformer are further optimized at the M06-2X/6-31+g(d,p) level of theory. Then, the single point energy calculations are performed at the M06-2X/6-311++G(3df,3pd) level of theory. RO radicals formed by the bimolecular reactions of $RO_2$ radicals with $HO_2$ radicals and NO also have multiple conformers. In order to obtain the lowest energy conformer, a similar methodology is employed in the present study.

Corresponding descriptions have been added in the page 6 line 152-168 of the revised manuscript:

*$RO_2$ radicals formed from the addition of $O_2$ to the carbon-centered site of alkyl radicals R have multiple possible conformers due to the different orientations of $O_2$ attack (Chen et al., 2021; Fu et al., 2020; Møller et al., 2016 and 2020). An initial structure of $RO_2$ radicals is optimized at the B3LYP/6-31+G(d) level and subsequently used as the starting geometry to perform the conformer search conducted using the Molclus program (Lu, 2024). The resulting structures are initially optimized at the B3LYP/6-31+G(d) level, as this method accurately predicts the relative energy ordering of different conformers (Møller et al., 2016 and 2020). For the intramolecular H-shift reactions of $RO_2$ radicals, the lengths of the O-O, C-H and O-H bonds in the conformational sampling of TSs are constrained to retain the cyclic TS structure. All unique conformers of R, TS and P within 5.0 kcal/mol with respect to the lowest energy conformer are*

*further optimized at the M06-2X/6-31+g(d,p) level of theory. Then, the single point energy calculations are performed at the M06-2X/6-311++G(3df,3pd) level of theory. RO radicals formed by the bimolecular reactions of $RO_2$ radicals with $HO_2$ radicals and NO also have multiple conformers. In order to obtain the lowest energy conformer, a similar methodology is employed in the present study.*

8. Lines 285-288: The authors use the consistency of the styrene- OH reaction with the toluene- OH reaction to justify discrepancies in calculated addition sites compared to previous literature. However, the distinct effects of methyl (-CH$_3$) and hydroperoxyl (-OOH) functional groups on the benzene ring should be explicitly discussed.

**Response:** Based on the Reviewer's suggestion, the relevant descriptions regarding the preferred OH-addition reaction have been added in the revised manuscript. For the reaction 1$^{st}$-ROOH (S4) + OH, the addition of OH radicals to either side of the benzene ring generates various alkyl radicals, as shown in Figure 2. In the present study, *syn*-OH-addition is defined as the scenario in which the addition of OH radicals occurs at the same side as the –OOH group, while *anti*-OH-addition is referred to the scenario in which the addition of OH radicals occurs at the opposite side as the –OOH group. For the *syn*-OH-addition reactions, the addition of OH radicals to the C1-site of 1$^{st}$-ROOH (S4) exhibits the lowest barrier ($\Delta E_a$ = 3.6 kcal/mol) due to the stability of the formed product, P$_{S4-add1'}$. A similar conclusion is also obtained from the *anti*-OH-addition reactions that the OH-addition pathway occurring at the C1-site is favorable ($\Delta E_a$ = 0.8 kcal/mol). Notably, the preferred OH-addition pathway in the *anti*-OH-addition reactions exhibits greater competitiveness compared to that in the *syn*-OH-addition reactions. It can be explained by the greater steric hindrance present in the latter reaction. In order to further evaluate the reliability of our results, $\Delta E_a$ of all the *syn*-OH-addition and *anti*-OH-addition reactions are recalculated using the DLPNO-CCSD(T)/aug-cc-pVTZ//M06-2X/6-311+G(d,p) method employed in the Zhang's study (Zhang et al., 2024). As shown in Table S3, the $\Delta E_a$ values obtained using the M06-2X/6-311++G(3df,3pd) method are in good agreement with those derived from the DLPNO-CCSD(T)/aug-cc-pVTZ method. The largest deviation and the average absolute deviation are 1.2 and 0.9 kcal/mol, respectively, indicating that the M06-2X/6-311++G(3df,3pd) method employed in this study is reliable. Based on the $\Delta E_a$ values obtained using the

DLPNO-CCSD(T)/aug-cc-pVTZ method, it can also be concluded that the addition of OH radicals to C1-site, occurring at the opposite direction relative to the –OOH group, is energetically favorable. The rate coefficients of the addition of OH radicals to the different sites of 1$^{st}$-ROOH are calculated to be $8.2 \times 10^{-12}$ (C1-site), $5.8 \times 10^{-15}$ (C2-site), $8.3 \times 10^{-15}$ (C3-site), $8.6 \times 10^{-15}$ (C4-site), $2.7 \times 10^{-12}$ (C5-site) and $4.1 \times 10^{-13}$ (C6-site) cm$^3$ molecule$^{-1}$ s$^{-1}$, respectively.

The aforementioned result is opposite to Zhang's finding that the addition of OH radicals to C6-site would be the most favorable pathway (Zhang et al., 2024). The discrepancy can be explained by the following three factors: (1) The 1$^{st}$-ROOH conformer selected in the Zhang's study is not the global minimum. In the present study, the global minimum conformer of 1$^{st}$-ROOH, identified through the conformer search, is found to be 2.2 kcal/mol lower than the 1$^{st}$-ROOH structure selected in the Zhang's study. (2) The pre-reactive complexes are not considered in the Zhang's study. The addition of OH radicals to C1-, C2-, C3- and C6-sites, occurring at the opposite direction relative to the –OOH group, are merely considered in the Zhang's study. They found that the apparent energy barrier of the addition of OH radicals to C6-site is smallest, and is therefore expected to be the favorable pathway. Actually, these OH-addition reactions are modulated by the pre-reactive complexes. It may be inappropriate to determine the favorable pathway based solely on apparent activation energy without considering the pre-reaction complexes. (3) From a geometric perspective, the addition of OH radicals to C6-site is associated with greater steric hindrance compared to other sites, as C6-atom connects with a larger functional group. Base on the aforementioned discussions, we believe that the addition of OH radicals to C6-site is unlikely to be the dominant pathway. Our calculations also confirm that the addition of OH radicals to C6-site is less importance compared to that at the C1-site.

Our conclusion is further supported by the reaction toluene + OH that the *ortho*-OH-addition reaction exhibits significant dominance, with the branching ratio of up to 69.8-75.8% (Wu et al., 2020; Zhang et al., 2019; Ji et al., 2017). Considering the high reactivity of *ortho*-OH-addition in the reactions toluene + OH and 1$^{st}$-ROOH (S4) + OH, the substitute effects of the –CH$_3$ and – OOH groups are explicitly discussed in the present study. Notably, the –CH$_3$ group in toluene is bonded to the C6 atom, and the –OOH group in 1$^{st}$-ROOH is bonded to the Cα atom, as depicted in Figure S5. The optimized geometries of toluene and 1$^{st}$-ROOH and the NPA atomic charges of

all the carbon atoms in the benzene ring are displayed in Figure S5. The C-C bond lengths and the C-C-C bond angles in the benzene ring of toluene are approximately 1.39 Å and 120°, respectively, which are consistent with those in the benzene ring of 1st-ROOH. The aforementioned results show the effect of the –CH₃ and –OOH groups on the geometric structure of benzene ring is negligible. From the perspective of NPA atomic charges, the charges on the C1 (-0.246 e) and C5 (-0.246 e) atoms are more greater than those on the other carbon atoms in the benzene ring of toluene. And the OH-adduct formed from the *ortho*-OH-addition reaction exhibits the greater stability. These results indicate that the –CH₃ group is a typical *ortho*-directing substituent and exerts an activating effect on the *ortho*-site of the benzene ring, which explains why the *ortho*-OH-addition reaction is predominant in the reaction toluene + OH. Compared with the charges on the carbon atoms in the benzene ring of toluene, the charges on C1 and C6 atoms increase by 0.013 e and 0.057 e, respectively, in 1st-ROOH, which can be attributed to the electron-withdrawing effect of the –OOH group. The charge on the C1 atom (-0.259 e) is the highest, and the stability of the resulting OH-adduct is the greatest, implying that the addition of OH radicals to C1-site is dominant in the reaction 1st-ROOH + OH. Therefore, a direct comparison of the favorable OH-addition pathway in the reactions toluene + OH and 1st-ROOH (S4) + OH is performed in this study.

[Figure]

**Figure 2.** PES for the oxidation of 1st-ROOH(S4) initiated by OH radicals at the M06-2X/6-311++G(3df,3pd)//M06-2X/6-31+g(d,p) level

**Table S3** The energy barriers ($\Delta E_a$ in kcal/mol) of the addition OH radicals to the different sites of 1st-ROOH (S4) calculated at the DLPNO-CCSD(T)/aug-cc-pVTZ//M06-2X/6-311+G(d,p) and M06-2X/6-311++G(3df,3pd)//M06-2X/6-31+g(d,p)levels

| (1ˢᵗ-ROOH (S4)) | DLPNO-CCSD(T)/aug-cc-pVTZ// M06-2X/6-311+G(d,p) | M06-2X/6-311++G(3df,3pd)// M06-2X/6-31+g(d,p) |
|---|---|---|
| *syn*-OH-addition ( OH is added in the same direction of -OOH group) | | |
| C1-site | 3.3 | 3.6 |
| C2-site | 7.8 | 8.8 |
| C3-site | 8.4 | 9.5 |
| C4-site | 5.4 | 5.7 |
| C5-site | 5.0 | 6.2 |
| C6-site | 4.7 | 5.8 |
| *anti*-OH-addition ( OH is added in the opposite direction of -OOH group) | | |
| C1-site | 2.0 | 0.8 |
| C2-site | 6.3 | 5.3 |
| C3-site | 6.7 | 6.0 |
| C4-site | 5.7 | 4.7 |
| C5-site | 2.6 | 1.4 |
| C6-site | 5.2 | 4.3 |

[Figure]

**Figure S5.** The geometric parameters of toluene and 1ˢᵗ-ROOH (S4) and the NPA atomic charges (blue font) of all the carbon atoms in the benzene ring predicted at the M06-2X/6-31+g(d,p) level

Corresponding descriptions have been added in the page 12 line 308-383 of the revised manuscript:

*The reaction 1ˢᵗ-ROOH (S4) + OH proceeds through the addition of OH radicals to either*

*side of the benzene ring to yield various alkyl radicals, as depicted in Figure 2. In the present study, syn-OH-addition is defined as the scenario in which the addition of OH radicals occurs at the same side as the –OOH group, while anti-OH-addition is referred to the scenario in which the addition of OH radicals occurs at the opposite side as the –OOH group. For the syn-OH-addition reactions, the addition of OH radicals to the C1-site of $1^{st}$-ROOH (S4) exhibits the lowest barrier ($\Delta E_a$ = 3.6 kcal/mol) due to the stability of the formed product, $P_{S4-add1'}$. A similar conclusion is also obtained from the anti-OH-addition reactions that the OH-addition pathway occurring at the C1-site is favorable ($\Delta E_a$ = 0.8 kcal/mol). Notably, the preferred OH-addition pathway in the anti-OH-addition reactions exhibits greater competitiveness compared to that in the syn-OH-addition reactions. It can be explained by the greater steric hindrance present in the latter reaction. In order to further evaluate the reliability of our results, $\Delta E_a$ of all the syn-OH-addition and anti-OH-addition reactions are recalculated using the DLPNO-CCSD(T)/ aug-cc-pVTZ//M06-2X/6-311+G(d,p) method. As shown in Table S3, the $\Delta E_a$ values obtained using the M06-2X/6-311++G(3df,3pd) method are in good agreement with those derived from the DLPNO-CCSD(T)/aug-cc-pVTZ method. The largest deviation and the average absolute deviation are 1.2 and 0.9 kcal/mol, respectively, indicating that the M06-2X/6-311++G(3df,3pd) method employed in this study is reliable. Based on the values of $\Delta E_a$ obtained using the DLPNO-CCSD(T)/ aug-cc-pVTZ method, it can also be concluded that the addition of OH radicals to C1-site, occurring at the opposite direction relative to the –OOH group, is energetically favorable. The rate coefficients of the addition of OH radicals to the different sites of $1^{st}$-ROOH are calculated to be 8.2 × $10^{-12}$ (C1-site), 5.8 × $10^{-15}$ (C2-site), 8.3 × $10^{-15}$ (C3-site), 8.6 × $10^{-15}$ (C4-site), 2.7 × $10^{-12}$ (C5-site) and 4.1 × $10^{-13}$ (C6-site) $cm^3$ $molecule^{-1}$ $s^{-1}$, respectively.*

*Our result is opposite to Zhang's finding that the addition of OH radicals to C6-site would be the most favorable pathway (Zhang et al., 2024). The discrepancy can be explained by the following three factors: (1) The $1^{st}$-ROOH conformer selected in the Zhang's study is not the global minimum. In the present study, the global minimum conformer of $1^{st}$-ROOH, identified through the conformer search, is found to be 2.2 kcal/mol lower than the $1^{st}$-ROOH structure selected in the Zhang's study. (2) The pre-reactive complexes are not considered in the Zhang's study. The addition of OH radicals to C1-, C2-, C3- and C6-sites, occurring at the opposite*

*direction relative to the –OOH group, are merely considered in the Zhang's study. They found that the apparent energy barrier of the addition of OH radicals to C6-site is smallest, and is therefore expected to be the favorable pathway. Actually, these OH-addition reactions are modulated by the pre-reactive complexes. It may be inappropriate to determine the favorable pathway based solely on apparent activation energy without considering the pre-reaction complexes. (3) From a geometric perspective, the addition of OH radicals to C6-site is associated with greater steric hindrance compared to other sites, as C6-atom connects with a larger functional group. Base on the aforementioned discussions, we believe that the addition of OH radicals to C6-site is unlikely to be the dominant pathway. Our calculations also confirm that the addition of OH radicals to C6-site is less importance compared to that at the C1-site.*

*Our conclusion is further supported by the reaction toluene + OH that the ortho-OH-addition reaction exhibits significant dominance, with the branching ratio of up to 69.8-75.8% (Ji et al., 2017; Zhang 2019; Wu et al., 2020). Considering the high reactivity of ortho-OH-addition in the reactions toluene + OH and $1^{st}$-ROOH (S4) + OH, the substitute effects of the –CH$_3$ and –OOH groups are explicitly discussed in the present study. Notably, the –CH$_3$ group in toluene is bonded to the C6 atom, and the –OOH group in $1^{st}$-ROOH is bonded to the Cα atom, as depicted in Figure S5. The optimized geometries of toluene and $1^{st}$-ROOH and the NPA atomic charges of all the carbon atoms in the benzene ring are displayed in Figure S5. The C-C bond lengths and the C-C-C bond angles in the benzene ring of toluene are approximately 1.39 Å and $120^{o}$, respectively, which are consistent with those in the benzene ring of $1^{st}$-ROOH. The aforementioned results show the effect of the –CH$_3$ and –OOH groups on the geometric structure of benzene ring is negligible. From the perspective of NPA atomic charges, the charges on the C1 (-0.246 e) and C5 (-0.246 e) atoms are more greater than those on the other carbon atoms in the benzene ring of toluene. And the OH-adduct formed from the ortho-OH-addition reaction exhibits the greater stability. These results indicate that the –CH$_3$ group is a typical ortho-directing substituent and exerts an activating effect on the ortho-site of the benzene ring, which explains why the ortho-OH-addition reaction is predominant in the reaction toluene + OH. Compared with the charges on the carbon atoms in the benzene ring of toluene, the charges on C1 and C6 atoms increase by 0.013 e and 0.057 e, respectively, in $1^{st}$-ROOH, which can be attributed to the electron-withdrawing effect of the –OOH group. The charge on the C1 atom (-0.259 e) is the*

*highest, and the stability of the resulting OH-adduct is the greatest, implying that the addition of OH radicals to C1-site is dominant in the reaction $1^{st}$-ROOH + OH. Therefore, a direct comparison of the favorable OH-addition pathway in the reactions toluene + OH and $1^{st}$-ROOH (S4) + OH is performed in this study.*

9. The manuscript contains multiple normative errors: (i) Hydroxyl radicals should be denoted as "OH radicals" or " OH"; (ii) $RO_2 \cdot$ and $RO \cdot$ should be consistently expressed as "$RO_2$ /RO " or "$RO_2$/RO radicals". The accuracy of scientific notation and writing requires improvement. Additionally, $\Delta E$ (e.g., in Fig. 5 and Supporting Information) should be presented in italics.

**Response:** Based on the Reviewer's suggestion, the hydroxyl radicals, $RO_2 \cdot$ and $RO \cdot$ have been denoted as OH radicals, and $RO_2$ radicals and RO radicals, respectively. The scientific notation and writing have been carefully corrected, and $\Delta E_a$ and $\Delta E_r$ have been revised in italics in the revised manuscript.

10. Manuscript nomenclature for intermediates/products is complex (such as PS4-add1'-OO-a/-s and PS4-add1"-OO-a-1) and poorly correlated with corresponding figures/structures, significantly hindering readability.

**Response:** Based on the Reviewer's suggestion, the nomenclature for intermediates and products have been simplified, and their structures have been corrected in the revised manuscript to improve clarity and readability.

11. In Lines 238-239 and 467-468, the author also quoted the literature when describing the rate constant. Is this rate constant calculated by the author or given in the literature?

**Response:** Based on the Reviewer's suggestion, the relevant descriptions regarding the pseudo-first-order rate constants have been added in the revised manuscript. In indoor environments, the concentration of $HO_2$ radicals is ~24 pptv, which is about half of the concentration of NO (Fu et al., 2024). Previous studies have reported that the rate coefficients $k_{RO2+HO2}$ and $k_{RO2+NO}$ for the reactions of alkyl peroxyl radicals with $HO_2$ radicals and NO are 1.7 $\times$ $10^{-11}$ and 9.0 $\times$ $10^{-12}$ $cm^3$ molecule$^{-1}$ s$^{-1}$ (Atkinson and Arey, 2003; Boyd et al., 2003),

respectively. The pseudo-first-order rate constants $k'_{RO2+HO2} = k_{RO2+HO2}$ [HO$_2$] and $k'_{RO2+NO} = k_{RO2+NO}$ [NO] are ~0.01 s$^{-1}$ in each case in indoor environments. In the atmosphere, the concentration of HO$_2$ radicals is 20-40 pptv (Wang et al., 2017; Bianchi et al., 2017), resulting in the pseudo-first-order rate constant $k'_{RO2+HO2} = k_{RO2+HO2}$ [HO$_2$] of 0.01-0.02 s$^{-1}$. The isomerization reaction of RO$_2$ radicals is competitive with the bimolecular reactions with HO$_2$ radicals only when the rate coefficient of intramolecular H-shifts exceeds 0.01-0.02 s$^{-1}$ in the low-NO$_x$ conditions. The typical atmospheric concentration of NO is 0.4-40 ppbv (Wang et al., 2017; Bianchi et al., 2017), leading to the pseudo-first-order rate constant $k'_{RO2+NO}$ of 0.1-10 s$^{-1}$. The intramolecular H-shift reaction of RO$_2$ radicals can compete with the bimolecular reaction with NO only when the rate coefficient of the former case exceeds 10 s$^{-1}$ in the high-NO$_x$ conditions. Similarly, for the association reactions of alkyl radicals with O$_2$, the rate coefficient $k_{R+O2}$ and atmospheric O$_2$ concentration are $6.0 \times 10^{-12}$ cm$^3$ molecule$^{-1}$ s$^{-1}$ and $5 \times 10^{18}$ molecule cm$^{-3}$, respectively (Ma et al., 2021), leading to the pseudo-first-order rate constant $k'_{R+O2} = k_{R+O2}$ [O$_2$] of $3.0 \times 10^7$ s$^{-1}$. The unimolecular decomposition of alkyl radicals is competitive only when their decay rate exceeds $3.0 \times 10^7$ s$^{-1}$.

Corresponding descriptions have been added in the page 9 line 253-269 and page 17 line 452-457 of the revised manuscript:

*Previous studies have reported that the rate coefficient $k_{RO2+HO2}$ for the reactions of alkyl peroxyl radicals with HO$_2$ radicals is $1.7 \times 10^{-11}$ cm$^3$ molecule$^{-1}$ s$^{-1}$ (Atkinson and Arey, 2003; Boyd et al., 2003). The typical atmospheric concentration of HO$_2$ radicals is 20-40 pptv (Wang et al., 2017; Bianchi et al., 2019), resulting in the pseudo-first-order rate constant $k'_{RO2+HO2} = k_{RO2+HO2}$ [HO$_2$] of 0.01-0.02 s$^{-1}$. The isomerization reaction of RO$_2$ radicals is competitive with the bimolecular reactions with HO$_2$ radicals only when the rate coefficient of intramolecular H-shifts exceeds 0.01-0.02 s$^{-1}$. In the high-NO$_x$ conditions, the bimolecular reaction of RO$_2$ radicals with NO is considered to be a dominant sink (Orlando and Tyndall, 2012; Vereecken et al., 2015). The rate coefficient $k_{RO2+NO}$ for the reaction of alkyl peroxyl radicals with NO is determined to be $9.0 \times 10^{-12}$ cm$^3$ molecule$^{-1}$ s$^{-1}$ (Atkinson and Arey, 2003; Bianchi et al., 2019). The typical atmospheric concentration of NO is 0.4-40 ppbv (Wang et al., 2017; Wang et al., 2019), leading to the pseudo-first-order rate constant $k'_{RO2+NO} = k_{RO2+NO}$ [NO] of 0.1-10 s$^{-1}$. The intramolecular H-shift reaction of RO$_2$ radicals can compete with the bimolecular reaction with NO only when the rate*

*coefficient of the former case exceeds 10 s$^{-1}$.*

*Adopting the rate coefficient $k_{R+O2}$ of $6.0 \times 10^{-12}$ cm$^3$ molecule$^{-1}$ s$^{-1}$ for the reactions of alkyl radicals with O$_2$, and the atmospheric O$_2$ concentration of $5 \times 10^{18}$ molecule cm$^{-3}$ (Ma et al., 2021), the pseudo-first-order rate constant $k'_{R+O2} = k_{R+O2}$ [O$_2$] is $3.0 \times 10^7$ s$^{-1}$. The unimolecular decomposition of alkyl radicals is competitive only when their decay rate exceeds $3.0 \times 10^7$ s$^{-1}$.*

12. Some figures are difficult to read due to formatting issues, e.g. In Figure 2, the font size is far too small. The energies and species labels are illegible at standard viewing sizes.

**Response:** Based on the Reviewer's suggestion, the font size, the energies and species labels in Figure 2 and 4 have been enlarged in the revised manuscript to improve clarity and readability.

[Figure]

**Figure 2.** PES for the oxidation of 1$^{st}$-ROOH(S4) initiated by OH radicals at the M06-2X/6-311++G(3df,3pd)//M06-2X/6-31+g(d,p) level

[Figure]

**Figure 4.** PES for the oxidation of $1^{st}$-$RONO_2$(S5) initiated by OH radicals at the M06-2X/6-311++G(3df,3pd)//M06-2X/6-31+g(d,p) level

[revised manuscript text omitted]